# LAYER-VARYING DEEP RESERVOIR COMPUTING ARCHITECTURE

## ABSTRACT

Data loss and corruption are common incidents that often lead to catastrophic consequences in both theoretical and experimental facets of data analytics. The aspiration to minimize the impacts of such consequences drives the demand for the development of effective data analytic tools and imputation methods to replace missing, corrupted, or artifacted data. The focus of this paper is on multivariate time series imputation, for which we develop a dynamical systems-theoretic deep learning approach. The central idea is to view a multivariate time series as a trajectory of a dynamical system. Then, we construct a deep reservoir computing architecture to model the temporal evolution of the system by using existing data in the time series. In particular, this architecture is composed of a cascade of echo state network (ESN) layers with diminishing reservoir sizes. We then propose a layer-by-layer training scheme, which gives rise to a deep learning-based time series imputation algorithm. We further provide a rigorous convergence analysis of this algorithm by exploiting the echo state property of ESN, and demonstrate the imputation performance as well as the efficiency of the training process by utilizing both synthetic and real-world datasets arising from diverse applications.

## 1 INTRODUCTION

With the rapid advancement in sensing and data storage technology, acquiring multivariate time series data has become easily achievable and increasingly popular across various fields, including medicine, economics, and climate science(Wu et al., 2017; Fernández-Gómez et al., 2017; Nguyen et al., 2020). Accompanying the abundance of data is a common occurrence of data irregularity, where partial recordings, such as sparse snapshots or multiple segments, are lost or damaged due to software corruption or hardware impairment. Missing data undoubtedly poses a significant challenge in data analytics tasks and may eventually lead to bottlenecks that hinder progress in scientific research. Driven by the critical need to minimize the impact of missing data, imputation—concerned with replacing missing data with appropriate substituted values—has garnered prominent attention in the realm of machine learning (ML).

The predominant focus of research in data imputation lies in crafting statistics-driven interpolation strategies and ML-enabled methods to deduce the values of absent data points, especially when those points are isolated, sparse, or randomly omitted. Yet, in the realm of multivariate time series, missing data can span consecutive (long) snapshots and multiple spatial dimensions. Imputing such data exceeds the capacities of many current techniques, necessitating the creation of novel methodologies.

**Our contributions.** In this work, we propose a novel dynamical systems-theoretic deep learning architecture for imputation of multivariate time series with isolated, sparse, or randomly omitted data samples as well as time series with missing data spanning consecutive (long) snapshots and multiple spatial dimensions. Specific contributions of this work are summarized below.

(1) We construct a deep reservoir computing network (RCN) architecture, composed of multiple layers of echo state network (ESN) layers with diminishing reservoir size to learn the underlying dynamical system generating the multivariate time series. (2) We propose a layer-by-layer training scheme for the proposed deep RCN, which gives rise to the DL-DRCN algorithm for multivariate time series imputation. (3) Rigorous convergence analyses of the DL-DRCN algorithm are provided.

(4) The high computational efficiency and performance of the algorithm are demonstrated by using both synthetic and real-world data.

## 2 RELATED WORKS

Missing value imputation problems have been widely studied over the past decades, with comprehensive insights provided in various works, such as (Schafer & Graham, 2002; Little & Rubin, 2002; Fang & Wang, 2020; Osman et al., 2018; Lin & Tsai, 2020; Emmanuel et al., 2021). The main body of works concerning adding and removing datapoints from time series focuses on regularizing irregularly sampled data. This refers to instances where the sampling rate is not uniform across the entire time series (Li & Marlin, 2020; Cao et al., 2018; Shukla & Marlin, 2021; Weerakody et al., 2021). Some of the regularizing techniques proposed in these works have also been tailored to tackle time-series imputation problems, where the missing data are filled in by mechanisms that adhere to certain assumptions on the distribution of the time series (Moritz et al., 2015; Kim et al., 2023; Little & Rubin, 2002; Azur et al., 2011; Fortuin et al., 2020). Interpolation is a popular non-probabilistic times-series imputation technique, e.g., linear interpolation (Shukla & Marlin, 2018; Junninen et al., 2004) and spline interpolation (Zhang, 2016; Junninen et al., 2004), where the values of missing data are estimated by fitting the observed data to linear and higher-order polynomials, respectively. However, these traditional methods often struggle to deliver satisfactory performance when applied to multivariate time series characterized by high spatial dimensions.

Due to their extraordinary computing power, various neural network-based multivariate time series imputation methods have been proposed, which have also achieved significant successes (Kidger et al., 2020; Che et al., 2018; Chen et al., 2018; Luo et al., 2018; Shan et al., 2021). These include methods that utilize static, deep generative models, such as generative adversarial network (GAN) (Gong et al., 2024; Luo et al., 2018), attention networks (Shukla & Marlin, 2021; Du et al., 2023), graph neural networks (Cini et al., 2022), and diffusion models (Tashiro et al., 2021; Alcaraz & Strodthoff, 2023; Rasul et al., 2021), as well as dynamic, recurrent neural networks (RNN) (Liu et al., 2019; Cao et al., 2018; Che et al., 2018; Yoon et al., 2019; Weerakody et al., 2021; Lipton et al., 2016). For instance, gated recurrent unit (GRU) models have gained in popularity in multivariate time-series imputation due to their capability to infer missing patterns and integrate them into observed data (Che et al., 2018; Cho et al., 2014; Chung et al., 2014; Kim et al., 2023). Typical examples of GRU-integrated imputation techniques include filling missing values with zero (GRU-zero), the mean of observed data (GRU-mean), the last observed values (GRU-forward), and an exponential decay mechanism between observations (GRU-D) (Che et al., 2018). Furthermore, a GAN-based framework is introduced, which employs a modified GRU to capture the complex dynamics of multivariate time series (Luo et al., 2018).

More recently, Neural Ordinary Differential Equation (Neural ODE) (Chen et al., 2018; Habiba & Pearlmutter, 2020) has been demonstrated as an efficient framework for the purpose of multivariate time series imputation. Different from traditional RNNs with discrete sequences of hidden layers, Neural ODEs are continuous-depth models whose outputs can be computed by solving ODEs. This characteristic of Neural Ordinary Differential Equations (ODEs) has been demonstrated to offer improved imputation performance and greater computational efficiency when compared to RNN-based methods. Further, ODE-RNNs have been introduced to further enhance the data imputation capabilities of Neural ODEs, exhibiting a notable advantage in handling irregularly sampled time series (Rubanova et al., 2019). Nevertheless, there is scarce literature addressing the imputation of multivariate time series with significant gaps, such as consecutive missing data spanning the temporal dimensions across various spatial dimensions. This gap persists despite increasing concern across diverse fields (Liu et al., 2020; Wu et al., 2022; Velasco-Gallego & Lazakis, 2022). In this work, we aim to address this gap by integrating techniques in dynamical systems theory with reservoir computing networks (RCNs), particularly echo state networks (ESNs) (Jaeger, 2007; 2005; Millea, 2014; Lukoševičius, 2012; Lu et al., 2017). We will begin by introducing ESNs from the perspective of dynamical systems theory in the next section and introduce a novel deep RCN architecture in section 4.1, which is fundamentally different from the deep ESN structures proposed in (Gallicchio et al., 2017; Jaeger, 2008) or other variations of RC frameworks as discussed in (Gauthier et al., 2021; Li et al., 2024).

## 3 BACKGROUND: DYNAMICAL SYSTEMS-THEORETIC TIME SERIES IMPUTATION

In this section, we carry out a dynamical systems-theoretic formulation of multivariate time series imputation. Then, we briefly review Echo State Networks (ESNs) from the perspective of dynamical systems, which are the learning models constituting the deep learning architecture for time series imputation proposed in this work.

### 3.1 MULTIVARIATE TIME SERIES IMPUTATION FROM DYNAMICAL SYSTEM VIEWPOINTS

**Notations.** We represent a multivariate time series in terms of a matrix $\boldsymbol{X}_{1:T} = [\boldsymbol{x}_1, \ldots, \boldsymbol{x}_T] \in \mathbb{R}^{d \times T}$, where the $t^{\text{th}}$ column $\boldsymbol{x}_t \in \mathbb{R}^d$ is the $t^{\text{th}}$ snapshot and the $i^{\text{th}}$ row $\boldsymbol{x}^i \in \mathbb{R}^{1 \times T}$ is the $i^{\text{th}}$ component of the mutivariate time series. We further assume that in each snapshot $\boldsymbol{x}_t$, at least one component is not missing. The imputation task considered in this work is to infer the missing values in some components of $\boldsymbol{X}_{1:T}$, denoted by $\boldsymbol{Y}_{1:T} = [\boldsymbol{y}_1, \ldots, \boldsymbol{y}_T] \in \mathbb{R}^{q \times T}$, by using the data in the complemented components $\boldsymbol{U}_{1:T} = [\boldsymbol{u}_1, \ldots, \boldsymbol{u}_T] \in \mathbb{R}^{p \times T}$ with $p = d - q$. We also allow $\boldsymbol{U}$ to contain missing data under the condition that $\boldsymbol{u}_t$ and $\boldsymbol{y}_t$ do not simultaneously have missing values for all $t = 1, \ldots, T$. We further associate the imputation target $\boldsymbol{Y}$ with mask $\boldsymbol{M}_{1:T} \in \mathbb{R}^{q \times T}$, whose $(i, j)$-entry is defied as $m_{ij} = 0$ if $y_{ij}$, the $(i, j)$-entry of $\boldsymbol{Y}_{1:T}$, is missing and $m_{ij} = 1$ otherwise.

**Missing scenarios.** The scenarios of missing data can be classified into two types: (1) data points are absent without discernible patterns, referred to as *random missing* scenario, which is similar to the case proposed in (Alcaraz & Strodthoff, 2023); (2) some components have missing data in consecutive snapshots, referred to *block missing*.

**Dynamical systems-theoretic approach to time series analytics.** The main idea is to think of the time series $\boldsymbol{Y}$, that is, the imputation target, is a trajectory of a dynamical system in the form of $\boldsymbol{y}_t = f(\boldsymbol{y}_{t-1}, \boldsymbol{u}_t)$. Of course, if the equation, specifically the function $f : \mathbb{R}^p \times \mathbb{R}^q \to \mathbb{R}^p$, modeling the system is discerned, the missing data in $\boldsymbol{Y}$ can be directly restored by evolving the system. From this perspective, the time series imputation task can be formulated as a dynamical system model learning task, for which recurrent neural networks (RNNs), owing to their dynamical system characteristic, stand out as the prime candidate for the learning tool.

### 3.2 DYNAMICAL SYSTEM REPRESENTATION OF ECHO STATE NETWORKS

Reservoir computing networks (RCNs) are a special class of RNNs, each of which contains a single hidden layer with no training parameter. The training-free hidden layer of an RCN is referred to as the **reservoir**, whose state, denoted as $\boldsymbol{r}_t \in \mathbb{R}^N$, is determined by the dynamical system $\boldsymbol{r}_t = \Psi_\Lambda(\boldsymbol{r}_{t-1}, \boldsymbol{u}_t)$, where $N$ is the reservoir dimension, equivalently the number of neurons in the hidden layer, $\boldsymbol{u}_t \in \mathbb{R}^p$ is the network input, and $\Psi_\Lambda : \mathbb{R}^N \times \mathbb{R}^p \to \mathbb{R}^N$ is called the **reservoir map**, parameterized by the set $\Lambda$ of the hyperparameters of the RCN, e.g., the reservori dimension $N$, and the reservoir weight matrices and bias vectors. In particular, in this work, we always pick $N < T$, the number of snapshots in the given time series. The output layer projects the reservoir state to a low-dimensional space $\mathbb{R}^q$ with $q \ll N$ by using the **readout map** $\hat{\boldsymbol{y}}_t = h(\boldsymbol{r}_t)$, which will be trained to minimize the discrepancy between the RCN output $\hat{\boldsymbol{y}}_t$ and the desired output $\boldsymbol{y}_t$. In general, the instantaneous discrepancy is evaluated by using a loss function $\mathcal{L}$ so that the cumulative discrepancy is given by $\ell(h; \mathcal{D}) = \sum_{t=1}^T \mathcal{L}(\hat{\boldsymbol{y}}_t, \boldsymbol{y}_t)$, where $\mathcal{D}$ is the given dataset and $T$ is the number of snapshots. In the training phase, the output layer weights are learned to obtain $h^* = \arg\min_h \ell(h; \mathcal{D})$.

**Echo state property** In the terminology of dynamical systems theory, echo state property (ESP) is essentially asymptotic stability of the reservoir system (Yildiz et al., 2012; Gonon & Ortega, 2021). Formally, given any input sequence $(\boldsymbol{u}_t)_{t \in \mathbb{N}}$ in $\mathbb{R}^p$, there exists a unique $\mathbf{r} \in \mathbb{R}^N$ such that $\boldsymbol{r}_t \to \mathbf{r}$ as $t \to \infty$. A sufficient condition to guarantee ESP is that the reservoir map $\Psi_\Lambda : \mathbb{R}^N \times \mathbb{R}^p \to \mathbb{R}^N$, given by, $(\mathbf{r}, \mathbf{u}) \mapsto \Psi_\Lambda(\mathbf{r}, \mathbf{u})$, is a contraction mapping in $\mathbf{r} \in \mathbb{R}^N$ and uniformly in $\mathbf{u} \in \mathbb{R}^p$, meaning, there is $L_\Psi < 1$ such that $\|\Psi_\Lambda(\mathbf{r}, \mathbf{u}) - \Psi_\Lambda(\mathbf{s}, \mathbf{u})\|$ for all $\mathbf{r}, \mathbf{s} \in \mathbb{R}^N$ and $\mathbf{u} \in \mathbb{R}^p$. Under this condition, given arbitrary initial reservoir states $\mathbf{r}_0, \mathbf{s}_0 \in \mathbb{R}^N$, $\|\boldsymbol{r}_t - \boldsymbol{s}_t\| = \|\Psi_\Lambda(\boldsymbol{r}_{t-1}, \boldsymbol{u}_t) - \Psi_\Lambda(\boldsymbol{s}_{t-1}, \boldsymbol{u}_t)\| < L_\Psi \|\boldsymbol{r}_{t-1} - \boldsymbol{s}_{t-1}\|$ holds so that $\|\boldsymbol{r}_t - \boldsymbol{s}_t\| < L_\Psi^t \|\boldsymbol{r}_0 - \boldsymbol{s}_0\|$ by induction, yielding $\|\boldsymbol{r}_t - \boldsymbol{s}_t\| \to 0$ as $t \to \infty$. Notice that the equilibrium state $\boldsymbol{r}_* = \lim_{t \to \infty} \boldsymbol{r}_t$ depends on the input sequence $(\boldsymbol{u}_t)_{t \in \mathbb{N}}$.

is a contraction mapping in the reservoir state (second) component, namely, $\|\Psi_\Lambda(\mathbf{r}, \mathbf{u}) - \Psi_\Lambda(\mathbf{s}, \mathbf{u})\| < \|\mathbf{r} - \mathbf{s}\|$ for any $\mathbf{r}, \mathbf{s} \in \mathbb{R}^N$ and $\mathbf{u} \in \mathbb{R}^p$. Then, for any two initial reservoir states $\mathbf{r}_0, \mathbf{s}_0 \in \mathbb{R}^N$, $\|\mathbf{r}_t - \mathbf{s}_t\| = \|\Psi_\Lambda(\mathbf{r}_{t-1}, \mathbf{u}_t) - \Psi_\Lambda(\mathbf{s}_{t-1}, \mathbf{u}_t)\| < \|\mathbf{r}_{t-1} - \mathbf{s}_{t-1}\|$ holds so that $(\|\mathbf{r}_t - \mathbf{s}_t\|)_{t \in \mathbb{N}}$ is a monotonically decreasing nonnegative sequence, and hence, necessarily converges to 0. Notice that the equilibrium state $\mathbf{r}$ depends on the choice of the input sequence $(\mathbf{u}_t)_{t \in \mathbb{N}}$.

In general, the hidden layer of an RCN has the structure of the composition of a linear network and an activation function. In this case, the reservoir map is given by $\Psi_\Lambda(\mathbf{r}, \mathbf{u}) = \psi(\boldsymbol{A}\mathbf{r} + \boldsymbol{B}\mathbf{u})$, where $A \in \mathbb{R}^{N \times N}$ and $B \in \mathbb{R}^{N \times p}$ are the hidden layer and input layer weight matrices, respectively, and $\psi : \mathbb{R} \to \mathbb{R}$ is the activation function acting on the vector $\boldsymbol{A}\mathbf{r} + \boldsymbol{B}\mathbf{u} \in \mathbb{R}^N$ component-wisely.

**Proposition 3.1.** *An RCN with the reservoir map given by $\Psi_\Lambda(\mathbf{r}, \mathbf{u}) = \psi(\boldsymbol{A}\mathbf{r} + \boldsymbol{B}\mathbf{u})$ has ESP if $\psi$ is Lipschitz continuous and $\|A\| < L^{-1}$, where $L$ is the Lipschitz constant of $\psi$ and $\|\cdot\|$ is the spectral norm of matrices, that is, the largest singular values.*

*Proof.* The main idea is to show $\|\Psi_\Lambda(\mathbf{r}, \mathbf{u}) - \Psi_\Lambda(\mathbf{s}, \mathbf{u})\| < L_\Psi \|\mathbf{r} - \mathbf{s}\|$ for any randomly chosen $\mathbf{r}, \mathbf{s} \in \mathbb{R}^N$ with some $L_\Psi > 0$, which follows directly from the fact that $\psi$ is Lipschitz continuous. See Appendix B.1 for more details. $\square$

The ESP constraint is crucial, as we require the reservoir states to not only converge asymptotically but also be independent of the choices of initial states and rely solely on the input. Therefore, to ensure the designed ESN possesses ESP property, we impose the constraints described in 3.1 on the designed reservoir mapping in the following section.

# 4 DEEP RESERVOIR COMPUTING ARCHITECTURE FOR MULTIVARIATE TIME SERIES ANALYTICS

In this section, we first propose the deep RCN architecture for multivariate time series analytics, which is composed of a cascade of ESN layers with diminishing reservoir dimensions and referred to as *DL-DRCN*. Then, we develop the multivariate time series imputation algorithm and prove its convergence.

## 4.1 MULTIVARIATE TIME SERIES IMPUTATION ALGORITHM

The central idea is to project the time series $\boldsymbol{Y}$ to the reservoir space determined by $\boldsymbol{U}$ in each ESN layer. This procedure results in a complete time series $\hat{\boldsymbol{Y}}$, which constitutes the learning reference for the successive ESN layer. The workflow of this DL-DRCN architecture is illustrated in Figure 1.

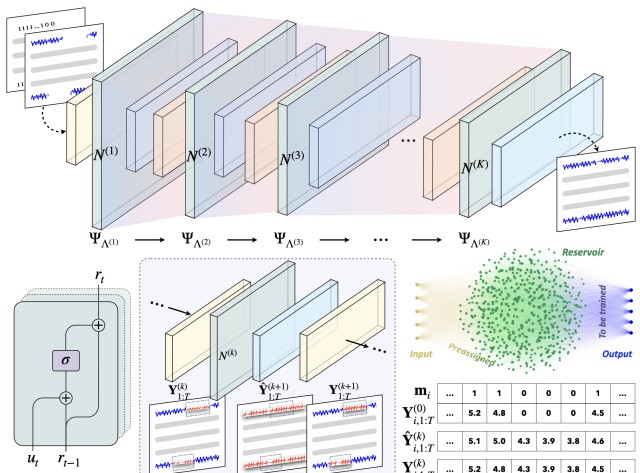

Figure 1: Illustration of the proposed DL-DRCN framework. The top panel demonstrates that the proposed model consists of multiple ESN layers, highlighting that the dimensions of ESN layers are decreasing layer-by-layer. The reservoir unit in each layer is demonstrated in the bottom left panel, where $u_t$ is the input to the reservoir unit at time $t$. The bottom middle panel showcases that, in the $k^{\text{th}}$ iteration of the algorithm (the $k^{\text{th}}$ ESN layer), the values of the missing datapoints in $\boldsymbol{Y}_{\omega:T}$ are substituted by those of the corresponding datapoints in the output $\hat{\boldsymbol{Y}}_{\omega:T}$ of the $k^{\text{th}}$ ESN layer. The structure of an ESN and the imputation process are depicted in the bottom right panel.

**DL-DRCN architecture.** The reservoir map $\Psi_{\Lambda^{(k)}} : \mathbb{R}^p \to \mathbb{R}^{N^{(k)}}$ of the $k^{\text{th}}$ layer of an ESN is chosen to be in the form of a leaky unit as

$$\boldsymbol{r}_t^{(k)} = (1 - \alpha^{(k)})\boldsymbol{r}_{t-1}^{(k)} + \alpha^{(k)}\sigma^{(k)}(\boldsymbol{A}^{(k)}\boldsymbol{r}_{t-1}^{(k)} + \boldsymbol{B}^{(k)}\boldsymbol{u}_t), \tag{1}$$

with the set of hyperparameters is given by $\Lambda^{(k)} = \{\boldsymbol{A}^{(k)}, \boldsymbol{B}^{(k)}, \sigma^{(k)}(\cdot), \alpha^{(k)}\}$, in which the input weight matrix $\boldsymbol{B}^{(k)} \in \mathbb{R}^{N^{(k)} \times p}$ is randomly chosen and the leakage rate $\alpha^{(k)} \in [0, 1]$ is picked to be

close to 1. To guarantee the ESP of this ESN layer, by Proposition 3.1, the condition $\|\boldsymbol{A}^{(k)}\| < 1/L^{(k)}$ with $L^{(k)}$ the Lipschitz constant of the activation function $\sigma^{(k)} : \mathbb{R} \to \mathbb{R}$ must be satisfied (see Appendix B.2 for more details). To construct such a weight matrix $\boldsymbol{A}^{(k)}$, we first pick a sparse matrix $\tilde{\boldsymbol{A}}^{(k)} \in \mathbb{R}^{N^{(k)} \times N^{(k)}}$ according to some preselected node degree $N_d$ (the number of nodes connected for each reservoir node), and then define $\boldsymbol{A}^{(k)} = \tilde{\boldsymbol{A}}^{(k)}/\|\tilde{\boldsymbol{A}}^{(k)}\|L^{(k)}$ (see Appendix H for more details). We then choose the readout map in the affine form $h^{(k)}(\boldsymbol{r}_t^{(k)}) = \tilde{\boldsymbol{C}}^{(k)}\boldsymbol{r}_t^{(k)} + \boldsymbol{b}^{(k)} = \boldsymbol{C}^{(k)}\tilde{\boldsymbol{r}}_t^{(k)}$, where the matrix $\boldsymbol{C}^{(k)} = \begin{bmatrix} \tilde{\boldsymbol{C}}^{(k)} & \boldsymbol{b}^{(k)} \end{bmatrix} \in \mathbb{R}^{q \times (N^{(k)}+1)}$, composed of the weight matrix and bias vector of the output layer, contains all the trainable parameters of this ESN layer, and $\tilde{\boldsymbol{r}}_t^{(k)} = \begin{bmatrix} \boldsymbol{r}_t^{(k)} \\ 1 \end{bmatrix}$. Note that during the training process only the "tails" are used instead of the entire reservoir state, which eliminates the impact of the choice of the initial reservoir state $\boldsymbol{r}_0^{(k)}$ according to ESP. Specifically, a washout length $1 < \omega < T$ is chosen in the way that $\boldsymbol{Y}_{1:(\omega-1)}^{(k)}$ does not have missing value and $T - \omega > N^{(k)}$. In particular, $\boldsymbol{C}^{(k)}$ is tuned to minimize $\ell(\boldsymbol{C}^{(k)}; \mathcal{D}^{(k)}) = \|\boldsymbol{C}^{(k)}\boldsymbol{R}_{\omega:T}^{(k)} - \boldsymbol{Y}_{\omega:T}^{(k)}\|_F^2 + \beta\|\boldsymbol{C}^{(k)}\|_F^2$, yielding

$$\boldsymbol{C}^{(k)^*} = \operatorname*{arg\,min}_{\boldsymbol{C}^{(k)} \in \mathbb{R}^{q \times N}} \ell(\boldsymbol{C}^{(k)}; \mathcal{D}^{(k)}) = \boldsymbol{Y}_{\omega:T}^{(k)}\boldsymbol{R}_{\omega:T}^{(k)^\top}\left(\boldsymbol{R}_{\omega:T}^{(k)}\boldsymbol{R}_{\omega:T}^{(k)^\top} + \beta\boldsymbol{I}\right)^{-1} := \boldsymbol{Y}_{\omega:T}^{(k)}\boldsymbol{R}_{\omega:T}^{(k)^\dagger}, \quad (2)$$

where $\boldsymbol{R}_{\omega:T}^{(k)} = \begin{bmatrix} \tilde{\boldsymbol{r}}_\omega^{(k)}, \dots, \tilde{\boldsymbol{r}}_T^{(k)} \end{bmatrix} \in \mathbb{R}^{(N^{(k)}+1) \times (T-\omega+1)}$ and $\mathcal{D}^{(k)} = \{\boldsymbol{U}_{1:T}, \boldsymbol{Y}_{1:T}^{(k)}\}$ with $\boldsymbol{Y}_{1:T}^{(k)}$ the imputed $\hat{\boldsymbol{Y}}_{1:T}$ obtained from the previous ESN layer.

**Layer-wise time series imputation.** The output of the $k^{\text{th}}$ ESN layer is then given by $\hat{\boldsymbol{y}}_t^{(k+1)} = \boldsymbol{C}^{(k)^*}\tilde{\boldsymbol{r}}_t^{(k)} = \boldsymbol{Y}_{\omega:T}^{(k)}\boldsymbol{R}_{\omega:T}^{(k)^\dagger}\tilde{\boldsymbol{r}}_t^{(k)}$. We then define the values of the missing datapoints in $\boldsymbol{Y}_{1:T}$ to be values of the corresponding datapoints, meaning those in the same snapshots and components, in $\hat{\boldsymbol{Y}}_{1:T}^{(k)}$. Computationally, this imputation procedure can be operated by using the mask matrix $\boldsymbol{M}$ as

$$\boldsymbol{y}_t^{(k+1)} = \boldsymbol{y}_t \odot \boldsymbol{m}_t + \hat{\boldsymbol{y}}_t^{(k+1)} \odot (\boldsymbol{1} - \boldsymbol{m}_t), \quad (3)$$

where $\odot$ denotes the Hadamard product. The imputed time series $\boldsymbol{Y}_{1:T}^{(k+1)} = \begin{bmatrix} \boldsymbol{y}_1^{(k+1)}, \dots, \boldsymbol{y}_T^{(k+1)} \end{bmatrix}$ is then used as the desired output to train the $(k+1)^{\text{th}}$ layer. This then gives rise to an iterative imputation algorithm, as shown in Algorithm 1, so that the imputation performance is improved layer-by-layer. The sequence of imputed time series $\boldsymbol{Y}_{1:T}^{(k)}$ resulting from this algorithm is guaranteed to converge to the groundtruth time series, which will be proved in the next section.

---

**Algorithm 1** DL-DRCN algorithm for multivariate time series imputation

**Input:** Dataset $\mathcal{D} = \{\boldsymbol{U}_{1:T}, \boldsymbol{Y}_{1:T}\}$, total layers number $K$
**Output:** Imputed result $\boldsymbol{Y}_{1:T}^{(K)}$

1: **while** $k < K$ **do**
2:     Generate $\Lambda^{(k)} = \{\boldsymbol{A}^{(k)}, \boldsymbol{B}^{(k)}, \sigma^{(k)}, \alpha^{(k)}\}$ and $\boldsymbol{r}_0^{(k)}$     ▷ Define ESN hyperparameters
3:     **for** $t = 1$ **to** $T$ **do**     ▷ Compute the evolution of the reservoir state
4:         $\boldsymbol{r}_t^{(k)} = (1 - \alpha^{(k)})\boldsymbol{r}_{t-1}^{(k)} + \alpha^{(k)}\sigma^{(k)}(\boldsymbol{A}^{(k)}\boldsymbol{r}_{t-1}^{(k)} + \boldsymbol{B}^{(k)}\boldsymbol{u}_t)$
5:     **end for**
6:     $\boldsymbol{C}^{(k)^*} = \boldsymbol{Y}_{\omega:T}^{(k)}\boldsymbol{R}_{\omega:T}^{(k)^\top}(\boldsymbol{R}_{\omega:T}^{(k)}\boldsymbol{R}_{\omega:T}^{(k)^\top} + \beta I)^{-1}$     ▷ Compute the output weight
7:     $\boldsymbol{Y}_{1:T}^{(k+1)} = \boldsymbol{Y}_{1:T}^{(0)} \odot \boldsymbol{m}_{1:T} + (\boldsymbol{C}^{(k)^*}\boldsymbol{R}_{1:T}^{(k)}) \odot (\boldsymbol{1} - \boldsymbol{m}_{1:T})$     ▷ Compute the update
8:     $k = k + 1$
9: **end while**
10: **Return:** $\boldsymbol{Y}_{1:T}^{(K)}$

---

## 4.2 Convergence Analysis of DL-DRCN

**RCN projection and reservoir space.** By representing the output output time series of the $k^{\text{th}}$ ESN layer in the matrix form as $\hat{\boldsymbol{Y}}_{\omega:T}^{(k+1)} = \boldsymbol{Y}_{\omega:T}^{(k)}\boldsymbol{R}_{\omega:T}^{(k)^\dagger}\boldsymbol{R}_{\omega:T}^{(k)}$, we observe that $\hat{\boldsymbol{Y}}_{\omega:T}^{(k+1)}$ is essentially

the projection of $\boldsymbol{Y}_{\omega:T}^{(k)}$, the time series resulting from the $k^{\text{th}}$ iteration of the proposed DL-DRCN algorithm, onto the space of reservior states, i.e., the vector space spanned by the row vectors $\boldsymbol{R}_{w:T}^{(k)}$.

**Convergence of DL-DRCN.** If the reservoir space of each ESN layer contains the groudtruth time series $\bar{\boldsymbol{Y}}_{\omega:T}$, then $\bar{\boldsymbol{Y}}_{\omega:T}$ necessarily satisfies the fixed point equation $\bar{\boldsymbol{Y}}_{\omega:T} = \bar{\boldsymbol{Y}}_{\omega:T}\boldsymbol{R}_{\omega:T}^{(k)}{}^{\dagger}\boldsymbol{R}_{\omega:T}^{(k)}$ for all $k$. From this perspective, Algorithm 1 presents a deep learning based iterative algorithm to solve this fixed point equation in the way that $\boldsymbol{Y}_{\omega:T}^{(k)} = \boldsymbol{Y}_{\omega:T}^{(k)}\boldsymbol{R}_{\omega:T}^{(k)}{}^{\dagger}\boldsymbol{R}_{\omega:T}^{(k)}$ as $k \to \infty$. This interpretation further motivates us to measure the deviation between the imputated and groundtruth time series by using the projection error $e^{(k)} = \|\boldsymbol{e}^{(k)}\| = \|\boldsymbol{Y}_{\omega:T}^{(k)} - \hat{\boldsymbol{Y}}_{\omega:T}^{(k+1)}\| = \|\boldsymbol{Y}_{\omega:T}^{(k)}(\boldsymbol{I} - \boldsymbol{R}_{\omega:T}^{(k)}{}^{\dagger}\boldsymbol{R}_{\omega:T}^{(k)})\|$.

In practice, an effective way to warrant $\bar{\boldsymbol{Y}}_{1:T}$ to be in the reservoir space of an ESN layer is to ensure that the number of neurons in the reservoir, equivalently, the dimension of the reservoir space, is large enough. However, higher dimension of the reservoir space leads to more expensive computational cost. To balance between the imputation performance and computational cost, we adopt the notion of minimal realization ESNs.

**Definition 4.1.** Given a fully observed time series $\boldsymbol{Y}_{1:T}$ and some positive number $\varepsilon > 0$, then an ESN of reservoir dimension $N_{\varepsilon}^{\boldsymbol{Y}}$ is said to be a **minimal $\varepsilon$-realization ESN for $\boldsymbol{Y}_{1:T}$** if $e = \|\boldsymbol{Y}_{1:T}(\boldsymbol{I} - \boldsymbol{R}_{1:T}{}^{\dagger}\boldsymbol{R}_{1:T})\| \le \varepsilon$ and there exists no other ESN of reservoir dimension $N < N_{\varepsilon}^{\boldsymbol{Y}}$ satisfying $e \le \varepsilon$. In addition, if $\varepsilon = 0$, then the ESN is called a **perfect realization ESN for $\boldsymbol{Y}_{1:T}$**.

**Theorem 4.2** (Convergence of DL-DRCN). *Given a multivariate time series $\mathcal{D} = \{\boldsymbol{U}_{1:T}, \boldsymbol{Y}_{1:T}\}$ and a projection error tolerance $\varepsilon > 0$. Let $\bar{\boldsymbol{Y}}_{1:T}$ be the groundtruth of $\boldsymbol{Y}_{1:T}$, then the sequence of time series $\boldsymbol{Y}_{1:T}^{(k)}$ imputing $\boldsymbol{Y}_{1:T}$ generated by Algorithm 1, using a deep reservoir computing network composed of multiple ESN layers with the reservoir dimensions greater than or equal to $N_{\varepsilon}^{\bar{\boldsymbol{Y}}}$, converges to a time series $\boldsymbol{Y}_{1:T}^{*}$ as $k \to \infty$ satisfying $\|\boldsymbol{Y}_{1:T}^{*} - \bar{\boldsymbol{Y}}_{1:T}\| < \delta$ for some $\delta$ depending on $\varepsilon$.*

*Proof.* The main idea is to show that the map $\boldsymbol{Y}_{1:T}^{(k)} \mapsto \boldsymbol{Y}_{1:T}^{(k)}\boldsymbol{R}_{1:T}^{(k)}{}^{\dagger}\boldsymbol{R}_{1:T}^{(k)}$ is a contraction mapping so that the sequence of projection errors $e^{(k)}$ is monotonically decreasing. This implies $\boldsymbol{e}^{(k)} \to 0$, and hence $\boldsymbol{Y}_{1:T} \to \boldsymbol{Y}_{1:T}^{*}$ for some time series $\boldsymbol{Y}_{1:T}^{*} \in \mathbb{R}^{p \times T}$ satisfying $\boldsymbol{Y}_{1:T}^{*} = \boldsymbol{Y}_{1:T}^{*}\boldsymbol{R}_{1:T}^{(k)}{}^{\dagger}\boldsymbol{R}_{1:T}^{(k)}$ for all $k$. See Appendix C for the details. $\square$

**Corollary 4.3.** *If each ESN layer in the deep reservoir computing network generating the imputation sequence $\boldsymbol{Y}_{1:T}^{(k)}$ is a perfect realization ESN for $\bar{\boldsymbol{Y}}_{1:T}$, then $\|\bar{\boldsymbol{Y}}_{1:T} - \boldsymbol{Y}_{1:T}^{(k)}\| \to 0$ as $k \to \infty$.*

*Proof.* The result direct follows from Theorem 4.2 by taking the projection error tolerance $\varepsilon = 0$. $\square$

In addition to the convergence guarantee of Algorithm 1, we also derive the convergence rate as follows.

**Proposition 4.4** (Convergence rate of DL-DRCN). *Given an error tolerance $\varepsilon > 0$, the projection error of the imputation time series satisfies $e^{(k)} < \varepsilon$ whenever $k > \frac{\ln\left(\epsilon - \sqrt{q}\delta\right) - \ln\left(E_{\boldsymbol{Y}} - \sqrt{q}\delta\right)}{\ln\left(\xi\right)}$, where $q$ is the dimension of the output time series, $\delta$ is the error bound defined in Theorem 4.2, $E_{\boldsymbol{Y}} = \sqrt{q}\max_{i=1}^{q}\|\boldsymbol{e}_i^{(0)}\|$, and $\xi = \|\boldsymbol{Y}_{1:T}\|/\|\bar{\boldsymbol{Y}}_{1:T}\| = \|\bar{\boldsymbol{Y}}_{1:T} \odot \boldsymbol{M}_{1:T}\|/\|\bar{\boldsymbol{Y}}_{1:T}\|$ with $\boldsymbol{M}_{1:T}$ being the mask of $\boldsymbol{Y}_{1:T}$.*

*Proof.* The result directly follows from the projection error analysis present in Theorem 4.2 with the given error tolerance. See Appendix D for the details. $\square$

**Reservoir dimension decreasing DL-DRCN.** Notice that the monotone decreasing property of the projection error sequence $\|\boldsymbol{e}^{(k)}\|$ indicates that the imputation performance is improved layer-by-layer. In other words, the imputation time series becomes closer to the limiting time series iteration-by-iteration. This observation enables us to decrease the dimension $N^{(k)}$ of the reservoir space of the $k^{\text{th}}$ ESN layer as $k$ increases without compromising the imputation performance, provided that $N^{(k)}$ is greater than to the dimension of the minimal $\varepsilon$-realization ESN. This is also an effective

way to improve the computational efficiency of the algorithm. Specifically, the total *floating point operations (flops)* counts of our DL-DRCN is $K(4N^2T + 2N^3 + 2dNT + 2ST + 3NT + 3qT + 2N)$ (with the detailed calculation of flops provided in Appendix G), or simply, $\mathcal{O}(KN^2T)$ flops. The main complexity of our algorithm comes from the matrix inverse operation, yielding a $\mathcal{O}(KN^2T)$ complexity with $N = N^{(1)}$ being the number of neurons in the first ESN layer and $K$ being the total number of ESN layers.

## 5 EXPERIMENTS

In this section, we demonstrate the performance and efficiency of the proposed DL-DRCN structure by using both synthetic and real-world datasets. Detailed descriptions of the used datasets are provided in Appendix I. In particular, the datasets with missing values were constructed by removing data points from the complete datasets according to predetermined mask matrices. The imputation performance were evaluated in terms of mean squared error (MSE), root mean squared error (RMSE), or mean absolute error (MAE) between the imputed values and groundtruth. Without further explanations, we always used the default choices of the hyperparameters of DL-DRCN illustrated in Appendix H. All simulations were run on a single GPU on an Apple M1 system with 16GB RAM.

### 5.1 SYNTHETIC DATA IMPUTATION

The synthetic dataset was generated by solving the Rössler System, a 3-dimensional ordinary differential equation (see Appendix I.1 for the detail).

**Imputation results.** In the construction of the datasets with missing values, we considered both of the block and random missing scenarios. Following the assumption of DL-DRCN presented in Section 3.1 that the input and imputation target do not simultaneously have missing values in any snapshot, without loss of generality, we let the first component $\boldsymbol{x}_{1:T}$ of the Rössler time series to be complete, which is used as the input of DL-DRCN, and the second component $\boldsymbol{y}_{1:T}$ be the imputation target. In particular, in each experiment, we randomly removed $p\%$ of datapoints from $\boldsymbol{y}_{1:T}$ for imputation, and then computed the mean square error (MSE) between the imputed values and the groundtruth. The missing ratio $p$ is chosen to be 10, 30, 50, and 70, and each experiment was run for 40 times. Table 1 shows the averaged MSE over the 40 runs of each experiment by using different imputation methods. We observe that DL-DRCN outperforms all baseline methods in the block missing cases and is also robust to the missing ratio.

**Time series inference.** In the case in which the missing data are in the snapshots towards the end of the time series, imputation coincides with inference (prediction). To showcase the applicability of DL-DRCN to this case, we inferred the last 70% of the datapoints in both $\boldsymbol{y}_{1:T}$ and $\boldsymbol{z}_{1:T}$, the third component of the Rössler time series. Figure 2 plots the groundtruth (blue) and DL-RCN inferred (red) values, from which we observe that the discrepancy between them is minor. This in turn

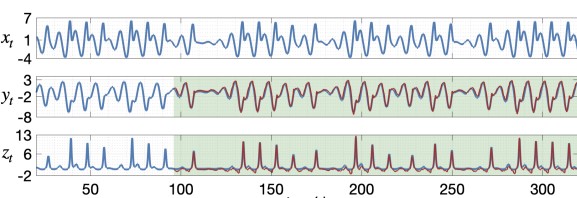

Figure 2: The inference result of the Rössler time series, which the inference time interval is indicated by the green block, and the groundtruth and inferred time series are plotted in blue and red, respectively.

expand the scope of DL-DRCN to focasting tasks in dynamical systems and time series analysis.

**Hyperparameter selections and ablation studies.** In addition to the comparison with baseline methods, we also use the synthetic Rössler time series to conduct hyperparameter selections and ablation studies of the proposed DL-DRCN. Similarly, we compute the averaged MSE between the groundtruth and imputed values over 40 runs of the DL-DRCN imputation for the $y$-component of the time series with random 50% block missing data.

The hyperparameters in DL-DRCN include the reservoir size ($N$), the activation function ($\sigma$), the leakage rate ($\alpha$), the regularization parameter ($\beta$), and the reservoir node degree ($N_d$), whose default values are $N = 500$, $\sigma = \tanh$, $\alpha = 0.8$, $\beta = 10^{-8}$, and $N_d = 300$. The results are shown in Figure

| Models | Block missing | | | | Random missing | | | |
|---|---|---|---|---|---|---|---|---|
| | 10% | 30% | 50% | 70% | 10% | 30% | 50% | 70% |
| L-ODE-RNN | 5.5391±1.2913 | 4.8986±0.3960 | 4.8848±0.2623 | 4.8909±0.1293 | 5.0634±0.2545 | 5.1027±0.1238 | 5.1480±0.2182 | 5.1611±0.4090 |
| Neural ODE | 5.6278±0.8260 | 4.7667±0.1639 | 4.9224±0.2004 | 4.8826±0.2393 | 5.053±0.411 | 5.1167±0.0930 | 5.1950±0.1070 | 5.1457±0.0634 |
| GRU-zero | 5.4321±1.2453 | 4.86±0.3962 | 4.8576±0.2483 | 4.8329±0.132 | 4.5291±0.3222 | 4.6207±0.3107 | 4.6779±0.2297 | 4.6967±0.2077 |
| GRU-mean | 6.4649±1.5812 | 5.3611±0.5092 | 5.0388±0.2782 | 4.8792±0.1472 | 1.0372±0.399 | 1.4415±0.5502 | 1.6429±0.5767 | 2.0858±0.5514 |
| GRU-D | 5.6112±1.3978 | 4.9461±0.4436 | 4.9503±0.2788 | 4.9193±0.149 | 3.6564±0.2716 | 3.7073±0.0940 | 3.7101±0.0847 | 3.7746±0.0421 |
| GP-VAE | 1.9044±0.0624 | 2.6999±0.5012 | 4.0845±1.2245 | 6.0013±1.0067 | 1.5057±0.0501 | 1.7922±0.5507 | 3.5998±1.0031 | 5.1105±1.5403 |
| SAITS | 4.9883±1.1233 | 4.94108±1.0760 | 5.7992±2.8246 | 5.2765±1.7186 | 2.1246±0.3111 | 2.3497±0.2234 | 2.7257±0.2856 | 3.1991±0.2083 |
| TimesNet | 6.2726±1.5298 | 5.5440±0.4867 | 5.2829±0.4752 | 4.9800±0.4095 | 0.4530±0.0905 | 0.5866±0.1320 | 1.0234±0.1394 | 2.7227±0.4164 |
| transformer | 4.9245±1.4567 | 4.3373±0.5615 | 4.3716±0.5022 | 4.5576±0.5580 | 4.2841±0.5683 | 4.5107±0.6466 | 4.4075±0.5729 | 4.5877±0.5336 |
| KNN | 5.3702±0.8923 | 5.2876±0.2499 | 5.8934±0.1618 | 7.0707±0.2493 | 4.9874±0.2522 | 5.3411±0.1644 | 5.8412±0.1234 | 6.5141±0.1341 |
| MICE | 5.4474±1.2441 | 4.8527±0.3979 | 4.8586±0.2367 | 4.8354±0.1166 | 5.0210±0.2410 | 5.0615±0.0851 | 5.0707±0.0665 | 5.0504±0.0480 |
| CubicSpline | 8.17±6.60 ($\times10^1$) | 5.88±7.02 ($\times10^2$) | 2.18±1.67 ($\times10^3$) | 3.66±3.05 ($\times10^3$) | **3.35±3.71** ($\times10^{-11}$) | **1.73±3.17** ($\times10^{-9}$) | **4.10±5.50** ($\times10^{-8}$) | **0.53±1.52** ($\times10^{-4}$) |
| Linear | 8.1378±2.9647 | 8.2769±3.5128 | 8.4306±2.9128 | 9.7103±5.7490 | 6.49±1.17 ($\times10^{-6}$) | 2.61±0.69 ($\times10^{-5}$) | 1.32±0.32 ($\times10^{-4}$) | 1.20±0.41 ($\times10^{-3}$) |
| vanilla ESN | 0.0589±0.0024 | 0.5351±0.0030 | 1.9140±0.0026 | 3.1112±0.0054 | 0.1497±0.0019 | 0.7324±0.0035 | 1.7280±0.0014 | 3.1157±0.0025 |
| DL-DRCN | **0.0569±0.0199** | **0.0512±0.0055** | **0.0539±0.0030** | **0.0626±0.0040** | 0.0472±0.0051 | 0.0492±0.0027 | 0.0522±0.0026 | 0.0573±0.0024 |

Table 1: Averaged mean squared error (mean ± std) between the groundtruth and imputed values over 40 runs of the Rössler time series imputation by using the proposed DL-DRCN and baseline methods. The best and second-best results are highlighted in **bold** and underlined. In particular, DL-DRCN outperforms all other baselines, especially in the block missing scenarios with high missing percentages. In addition, the averaged MSE of DL-DRCN decreases mildly with respect to the increase of the missing percentage, showing the robustness of DL-DRCN to the amount of missing data.

3, in which each subfigure plots the averaged MES versus the layer (iteration) number with different selections of one hyperparameter and others fixed. We observe that distinct hyperparameter values do not lead to drastically different imputation performance, and quantitatively, the averaged MSE stays in the same order in all the cases. Moreover, the algorithm always converges in no more than 10 iterations. These indeed demonstrate the stability, with respect to hyperparameter values, and fast convergence of DL-DRCN. In the ablation studies, we investigate the impact of the bias vectors in the ESN output layers and the results are shown in Figure 3 (f).

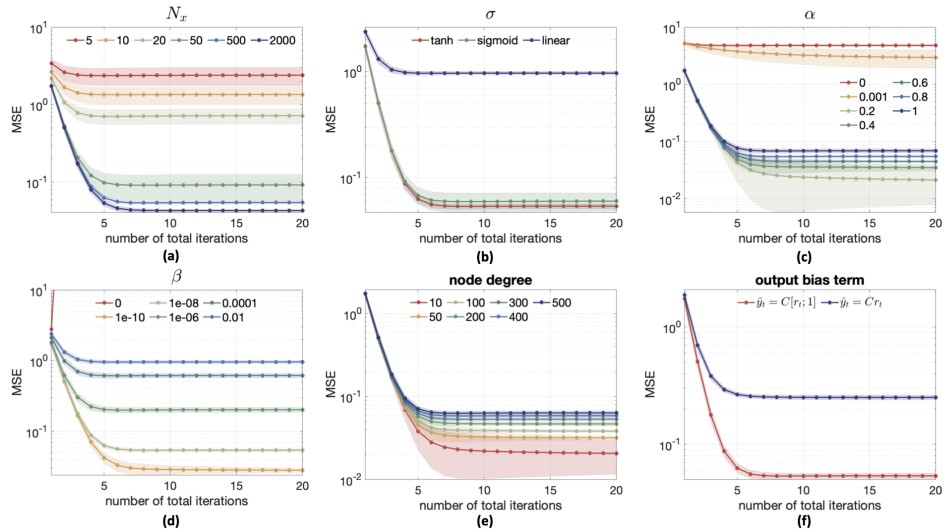

Figure 3: The comparisons of various hyperparameter selections, including the reservoir size ($N_x$), the activation function ($\sigma$), the leakage rate ($\alpha$), the regularization parameter ($\beta$), and the reservoir node degree, are shown in panels (a)-(e); while the results of ablation studies assessing the impact of the output bias term is present in (f).

## 5.2 REAL WORLD DATA IMPUTATION

**Gesture phase segmentation dataset (Gesture).** We considered the gesture phase segmentation dataset from (Madeo et al., 2014), which consists of regularly recorded video data with no missing values (see Appendix I.3 for the detailed description of this dataset). We extracted 32 time series from the provided processed file, each of which characterizes the velocity or acceleration of the hand or wrist movement over time. We kept 7 of them complete, which then used as the DL-RCN input, and then randomly removed $\mathfrak{p}\%$ of the data in each of the remaining time series by using both of the block and random missing schemes with $\mathfrak{p} = 30, 50, 70$. Similar to the synthetic dataset case, we computed the averaged MSE between the groundturth and imputed values over 40 runs of each experiment, and the imputation results are reported in Table 2. It can be observed that the proposed DL-DRCN

significantly outperforms almost all the existing imputation methods applied to this dataset with different missing schemes and percentages. To be more specific, in the case of randomly missing 30% datapoints, the averaged MSE resulting from DL-DRCL is about 50% of those resulting from other methods, demonstrating the practical applicability and excellent performance of the proposed methods.

**Imputation for medical datasets.** Applications of machine learning and artificial intelligence techniques to medical fields are emerging and prosperous research topics in this decade. In this example, we demonstrate the applicability of the proposed DL-DRCN to medical datasets, including the *PTB-XL Electrocardiography (ECG) Dataset* (Goldberger et al., 2000) and the clinical intensive care unit (ICU) dataset from the *PhysioNet Challenge 2012* (Silva et al., 2012).

For the PTB-XL ECG Dataset, we preprocessed it to a mutivariate time series composed of 12-lead ECG signals (12 components) with 1000 snapshots by using the method presented in (Alcaraz & Strodthoff, 2023). For each component, we randomly drop 20% of the datapoints by carrying out the block missing scenario, and then applied DL-DRCN to impute the constructed multivariate time series with missing data component-by-component. Specifically, for each component to be imputed, we pick another component, whose missing data are located at different snapshots as this one, as the input to DL-DRCN, and then computed the averaged MSE over 5 runs of the imputation. The results are shown in the right column of Table 3 as well as in Figure 4.

The ICU dataset, obtained from the MIMIC II database, comprises 4000 multivariate time series documenting up to 41 vital signs of 4000 patients measured over 2-hour periods. We first applied the preprocessing method presented in (Tashiro et al., 2021; Che et al., 2018) to select 35 key vitals and eliminate the patients' data with less than 5 snapshots, which results 3980 (patients) multivariate time series of 35 components (vitals). For each patient, we used the same removal strategy with 10% missing percent and imputation procedure as the ECG dataset to compute the averaged MSE. Then, we took the mean of the resulting averaged MSE over the 3980 patients as the imputation performance measure, which are shown in the left column of Table 3.

**Imputation results for real datasets.** We summarize the imputation results of Gesture in Table 2 and both ECG and Physionet datasets in Table 3, with the best results highlighted in **bold**. The results with asterisks are cited from the original paper. In the Gesture dataset, our model consistently outperforms them in all scenarios when the missing percentage exceeds 50% compared to other baseline methods. In the ECG dataset, our model perform similarly to the state-of-the-art imputation and is capable of capturing dynamic patterns (the peaks of all 12 signals) as shown in Figure 4. In the Physionet dataset, since the assumption described in section 4.1 is not necessarily satisfied, we first applied linear interpolation to those time points before imputation. The imputation results demonstrate our model offers consistent performance across both datasets and provides meaningful imputation in the block missing scenario.

## 6 CONCLUSION

In this paper, we present a novel RCN-based method for analyzing multivariate time series data. By utilizing the dynamical system characterization of RCNs, we propose a deep RCN architecture, composed of a sequence of ESN layers, to learn the systems generating the data so that the missing data can be recovered by the temporal evolution of the learned systems. To improve the computational efficiency, the reservoir size of an ESN layer is decreased compared to that of its predecessor, and the deep RCN is trained layer-by-layer, which gives rise to an iterative multivariate time series imputation algorithm. We further provide the computational analysis as well as the convergence proof of the algorithm, and demonstrate its excellent performance by using both synthetic and real-world data.

**Limitations and future works.** Since the ESP is required to be satisfied while training the proposed DL-DRCN to eliminate the impact of the choices of the initial reservoir states on the imputation result, the proposed algorithm may not perform satisfactorily if the missing data are located at the beginning of the time series. On the other hand, the convergence of the algorithm requires that the dimension of the reservoir state in each ESN layer is set to be greater than a lower bound (the dimension of minimal realization ESN). Since this requirement is highly dependent on data and varies across different scenarios, this design parameter often relies on empirical experiments.

In this work, the deep RCN is trained by using the observed components of multivariate time series. The model requires the input and the imputed target sequences to contain no overlapping time slots where data is missing. In principle, the algorithm is still applicable if there are overlapping missing data in both input and target time series. This will be investigated in our future works. In addition, the requirement mentioned above, that is, the reservoir dimension is greater than the spatial dimension of the time series is primarily due to the use of linear ESN output layers for projecting the desired outputs to the reservoir spaces. In the future, we will design ESNs with nonlinear readout maps in the output layers to waive this requirement, which gives another approach to improve the computational efficiency of the algorithm.

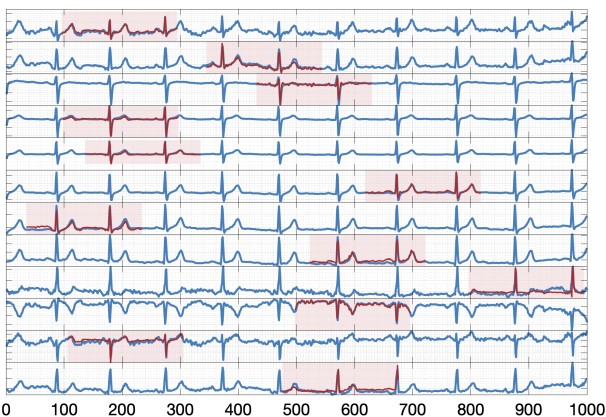

Figure 4: The imputation result of a block missing scenario for ECG dataset. The missing blocks (highlighted in red) demonstrate the reconstruction results of applying DL-DRCN model to fill in missing values, where the reconstructed trajectories and the truth trajectories are highlighted in blue and red, respectively.

|  | Block missing | | |
|---|---|---|---|
| Models | 30% | 50% | 70% |
| L-ODE-RNN | 11.97±4.79 | 13.97±16.42 | 14.42±12.43 |
| GRU-zero | 4.67±0.56 | 5.05±0.52 | 4.87±0.86 |
| GRU-mean | 5.05±0.58 | 4.85±0.89 | 5.13±0.85 |
| GRU-D | **4.13±0.87** | 4.60±1.21 | 5.34±2.72 |
| KNN | 9.61±2.20 | 8.77±1.03 | 8.98±0.42 |
| DL-DRCN | 4.732±1.060 | **4.003±0.525** | **4.709±0.648** |
|  | Random missing | | |
| Models | 30% | 50% | 70% |
| L-ODE-RNN | 13.15±5.06 | 16.36±8.60 | 17.64±7.52 |
| GRU-zero | 4.76±0.244 | 4±0.297 | 4.89±0.430 |
| GRU-mean | 4.79±0.27 | 4.88±0.37 | 5.01±0.48 |
| GRU-D | 4.11±0.46 | 4.09±0.47 | 4.29±0.71 |
| KNN | 8.92±0.78 | 9.11±0.63 | 9.51±0.50 |
| DL-DRCN | **2.866±0.435** | **3.154±0.395** | **3.460±0.357** |

Table 2: MSE (mean ± std) ($\times 10^{-5}$) of imputation results for the Gesture dataset.

| Models | Physionet | ECG |
|---|---|---|
| Linear | 0.615±0.056 | 0.266±0.004 |
| KNN | 0.662±0.056 | 0.232±0.002 |
| CSDI/SSSD | 0.217±0.001* | **0.023±9e-4*** |
| DL-DRCN | **0.103±0.001** | 0.055±7e-3 |

Table 3: MAE/RMSE (mean ± std) of imputation results for PhysioNet/ECG datasets.

## POTENTIAL BROADER IMPACT

This paper presents a deep reservoir computing architecture and an associated training algorithm for missing data imputation in multivariate time series. The method itself is broadly applicable to diverse domains where artifacted time series data are abundant. For instance, data lost due to measurement or process noise, sensor failure or downtime, and other environmental factors can cause missing data or data loss, and the proposed approach is applicable in all these scenarios to recover the time series.

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
