## A  Norm Bound Conditions

**Lemma A.1.** *Let the largest singular values of $\boldsymbol{R}_{\omega:T}^{(k)}$ be $s^{(k)}$, then the norm of $\boldsymbol{H}^{(k)} = \boldsymbol{R}_{\omega:T}^{(k)}{}^{\dagger} \boldsymbol{R}_{\omega:T}^{(k)}$ satisfies $\|\boldsymbol{H}^{(k)}\| = \|\boldsymbol{R}_{\omega:T}^{(k)}{}^{\dagger} \boldsymbol{R}_{\omega:T}^{(k)}\| = s^{(k)^2}/(s^{(k)^2} + \beta) \in (0,1)$.*

*Proof.* Let $\boldsymbol{R}_{\omega:T}^{(k)} = \boldsymbol{U}\boldsymbol{\Sigma}\boldsymbol{V}^{\top}$ be the singular value decomposition of $\boldsymbol{R}_{\omega:T}^{(k)}$ (Superscripts of $\boldsymbol{U}^{(k)}$, $\boldsymbol{\Sigma}^{(k)}$, and $\boldsymbol{V}^{(k)}$ are neglected in this proof to avoid cumbersome notations). This together with $\boldsymbol{R}_{\omega:T}^{(k)}{}^{\dagger} = \boldsymbol{R}_{\omega:T}^{(k)}{}^{\top}(\boldsymbol{R}_{\omega:T}^{(k)}\boldsymbol{R}_{\omega:T}^{(k)}{}^{\top} + \beta\boldsymbol{I})^{-1}$ from (2) yields

$$
\begin{aligned}
\boldsymbol{R}_{\omega:T}^{(k)}{}^{\dagger} \boldsymbol{R}_{\omega:T}^{(k)} &= \boldsymbol{R}_{\omega:T}^{(k)}{}^{\top}(\boldsymbol{R}_{\omega:T}^{(k)}\boldsymbol{R}_{\omega:T}^{(k)}{}^{\top} + \beta\boldsymbol{I})^{-1}\boldsymbol{R}_{\omega:T}^{(k)} \\
&= \boldsymbol{V}\boldsymbol{\Sigma}^{\top}\boldsymbol{U}^{\top}\left(\boldsymbol{U}\boldsymbol{\Sigma}\boldsymbol{V}^{\top}\boldsymbol{V}\boldsymbol{\Sigma}^{\top}\boldsymbol{U}^{\top} + \beta\boldsymbol{I}\right)^{-1}\boldsymbol{U}\boldsymbol{\Sigma}\boldsymbol{V}^{\top} \\
&= \boldsymbol{V}\boldsymbol{\Sigma}^{\top}\boldsymbol{U}^{\top}\left(\boldsymbol{U}\left(\boldsymbol{\Sigma}\boldsymbol{\Sigma}^{\top} + \beta\boldsymbol{I}\right)\boldsymbol{U}^{\top}\right)^{-1}\boldsymbol{U}\boldsymbol{\Sigma}\boldsymbol{V}^{\top} \\
&= \boldsymbol{V}\boldsymbol{\Sigma}^{\top}\boldsymbol{U}^{\top}\boldsymbol{U}\left(\boldsymbol{\Sigma}\boldsymbol{\Sigma}^{\top} + \beta\boldsymbol{I}\right)^{-1}\boldsymbol{U}^{\top}\boldsymbol{U}\boldsymbol{\Sigma}\boldsymbol{V}^{\top} \\
&= \boldsymbol{V}\boldsymbol{\Sigma}^{\top}\left(\boldsymbol{\Sigma}\boldsymbol{\Sigma}^{\top} + \beta\boldsymbol{I}\right)^{-1}\boldsymbol{\Sigma}\boldsymbol{V}^{\top}.
\end{aligned}
$$

As a result, $\|\boldsymbol{H}^{(k)}\| = \|\boldsymbol{R}_{\omega:T}^{(k)}{}^{\dagger} \boldsymbol{R}_{\omega:T}^{(k)}\| = \|\boldsymbol{\Sigma}^{\top}\left(\boldsymbol{\Sigma}\boldsymbol{\Sigma}^{\top} + \beta\boldsymbol{I}\right)^{-1}\boldsymbol{\Sigma}\| = s^{(k)^2}/(s^{(k)^2} + \beta) \in (0,1)$. $\square$

**Lemma A.2.** *Let $\boldsymbol{Y}_{1:T}$ be an time series and $\boldsymbol{m}_{1:T}$ be the associated mask. Assume $\boldsymbol{Y}_{1:T} \odot \boldsymbol{m}_{1:T} \neq \boldsymbol{Y}_{1:T}$, then $\|\boldsymbol{Y}_{1:T} \odot \boldsymbol{m}_{1:T}\| < \|\boldsymbol{Y}_{1:T}\|$. Equivalently, we have $\|\boldsymbol{Y}_{1:T} \odot \boldsymbol{m}_{1:T}\| = \xi\|\boldsymbol{Y}_{1:T}\|$ with $\xi < 1$. Here, $\|\cdot\|$ denotes the $\ell^2$ norm (Euclidean norm) and $\odot$ denotes the Hadamard product (elementwise product) of two vectors.*

*Proof.* Clearly, we have $\|\boldsymbol{Y}_{1:T} \odot \boldsymbol{m}_{1:T}\| = \sum_{t=1}^{T} y_t^2 m_t \leq \sum_{t=1}^{T} y_t^2 = \|\boldsymbol{Y}_{1:T}\|$ with the equality holds if and only if $\boldsymbol{Y}_{1:T} \odot \boldsymbol{m}_{1:T} = \boldsymbol{Y}_{1:T}$ or $\boldsymbol{Y}_{1:T} = \boldsymbol{0}$ (this case is excluded by default). $\square$

## B  Lipschitz Conditions of reservoir maps

**Proposition B.1.** *An RCN with the reservoir map given by $\Psi_{\Lambda}(\boldsymbol{r}, \boldsymbol{u}) = \psi(\boldsymbol{Ar} + \boldsymbol{Bu})$ has ESP if $\psi$ is Lipschitz continuous and $\|A\| < L^{-1}$, where $L$ is the Lipschitz constant of $\psi$ and $\|\cdot\|$ is the spectral norm of matrices, that is, the largest singular values.*

*Proof.* Pick any $\mathbf{r}, \mathbf{s} \in \mathbb{R}^N$ and $\mathbf{u} \in \mathbb{R}^p$, the Lipschitz continuity of $\psi$ gives $\|\Psi_{\Lambda}(\boldsymbol{r}, \boldsymbol{u}) - \Psi_{\Lambda}(\boldsymbol{s}, \boldsymbol{u})\| = \|\psi(\boldsymbol{Ar} + \boldsymbol{Bu}) - \psi(\boldsymbol{As} + \boldsymbol{Bu})\| \leq L\|\boldsymbol{A}(\boldsymbol{r} - \boldsymbol{s})\| \leq L\|\boldsymbol{A}\|\|\boldsymbol{r} - \boldsymbol{s}\| < \|\boldsymbol{r} - \boldsymbol{s}\|$. This implies that $\Psi_{\Lambda}$ is a contraction mapping, and hence the RCN satisfies ESP. $\square$

**Lemma B.2.** *Let $\alpha \in (0, 1]$ and $\sigma(\cdot)$ be a Lipschitz continuous activation function with Lipschitz constant $L$. Then, an RCN with the reservoir map given by (1), i.e., $\boldsymbol{r}_t = (1 - \alpha)\boldsymbol{r}_{t-1} + \alpha\sigma(\boldsymbol{Ar}_{t-1} + \boldsymbol{Bu}_t)$, has ESP if $\|A\| < L^{-1}$.*

*Proof.* Pick an arbitrary input sequence $\mathbf{u}_t \in \mathbb{R}^p$ and consider any two distinct initial reservoir state $\boldsymbol{r}_0, \boldsymbol{s}_0 \in \mathbb{R}^N$, then $\|\boldsymbol{r}_t - \boldsymbol{s}_t\| \leq (1 - \alpha)\|\boldsymbol{r}_{t-1} - \boldsymbol{s}_{t-1}\| + \alpha\|\sigma(\boldsymbol{Ar}_{t-1} + \boldsymbol{Bu}_t) - \sigma(\boldsymbol{As}_{t-1} + \boldsymbol{Bu}_t)\| \leq \left[(1 - \alpha) + \alpha L\|\boldsymbol{A}\|\right]\|\boldsymbol{r}_{t-1} - \boldsymbol{s}_{t-1}\|$. Further defining $k_A := (1 - \alpha) + \alpha L\|\boldsymbol{A}\|$ leads to $\|\boldsymbol{r}_t - \boldsymbol{s}_t\| \leq k_A\|\boldsymbol{r}_{t-1} - \boldsymbol{s}_{t-1}\|$, which implies by choosing $\|A\| < L^{-1}$, the sequence $\boldsymbol{r}_t$ converges to the sequence $\boldsymbol{s}_t$ as desired. $\square$

## C  Convergence guarantee of DL-DRCN

**Theorem C.1** (Convergence of **DL-DRCN**). *Given a multivariate time series $\mathcal{D} = \{\boldsymbol{U}_{1:T}, \boldsymbol{Y}_{1:T}\}$ and a projection error tolerance $\varepsilon > 0$. Let $\bar{\boldsymbol{Y}}_{1:T}$ be the groundtruth of $\boldsymbol{Y}_{1:T}$, then the sequence*

*of time series $\boldsymbol{Y}_{1:T}^{(k)}$ imputing $\boldsymbol{Y}_{1:T}$ generated by Algorithm 1, using a deep reservoir computing network composed of multiple ESN layers with the reservoir dimensions greater than or equal to $N_\varepsilon^{\bar{Y}}$, converges to a time series $\boldsymbol{Y}_{1:T}^*$ as $k \to \infty$ satisfying $\|\boldsymbol{Y}_{1:T}^* - \bar{\boldsymbol{Y}}_{1:T}\| < \delta$ for some $\delta$ depending on $\varepsilon$.*

*Proof.* Let

$$\boldsymbol{e}^{(k)} = \boldsymbol{Y}_{\omega:T}^{(k)} - \bar{\boldsymbol{Y}}_{\omega:T} \tag{4}$$

denote the error between the imputed time series $\boldsymbol{Y}_{1:T}^{(k)}$ and the ground truth $\bar{\boldsymbol{Y}}_{1:T}$ after the completion of layer $k$ of the DL-DRCN, then we will show that $\|\boldsymbol{e}^{(k)}\|$ is bounded by $d_\varepsilon$ as $k \to \infty$.

First, we denote the projection of $\bar{\boldsymbol{Y}}_{1:T}$ onto the reservoir space in $k^{\text{th}}$ ESN layer as $\hat{\bar{\boldsymbol{Y}}}_{\omega:T}^{(k)}$, meaning

$$\hat{\bar{\boldsymbol{Y}}}_{\omega:T}^{(k)} = \bar{\boldsymbol{Y}}_{\omega:T}\, \boldsymbol{R}_{\omega:T}^{(k)}{}^\dagger \boldsymbol{R}_{\omega:T}^{(k)}. \tag{5}$$

As a result, in each iteration, the $i^{\text{th}}$ row of the error matrix $\boldsymbol{e}^{(k)}$ satisfies

$$\begin{aligned}
\boldsymbol{e}_i^{(k)} &= \left(\hat{\boldsymbol{Y}}_{i,\omega:T}^{(k)} - \bar{\boldsymbol{Y}}_{i,\omega:T}\right) \operatorname{diag}(\mathbf{1}^\top - \boldsymbol{m}_i) \\
&= \left[\boldsymbol{Y}_{i,\omega:T}^{(k-1)} \boldsymbol{R}_{\omega:T}^{(k-1)\dagger} \boldsymbol{R}_{\omega:T}^{(k-1)} - \bar{\boldsymbol{Y}}_{i,\omega:T}\right] \operatorname{diag}(\mathbf{1}^\top - \boldsymbol{m}_i) \\
&= \left[\boldsymbol{Y}_{i,\omega:T}^{(k-1)} - \bar{\boldsymbol{Y}}_{i,\omega:T}\right] \boldsymbol{H}^{(k-1)} \operatorname{diag}(\mathbf{1}^\top - \boldsymbol{m}_i) + \left[\hat{\bar{\boldsymbol{Y}}}_{i,\omega:T}^{(k-1)} - \bar{\boldsymbol{Y}}_{i,\omega:T}\right] \operatorname{diag}(\mathbf{1}^\top - \boldsymbol{m}_i) \\
&= \boldsymbol{e}_i^{(k-1)} \boldsymbol{H}^{(k-1)} \operatorname{diag}(\mathbf{1}^\top - \boldsymbol{m}_i) + \varepsilon_{\bar{\boldsymbol{Y}}}^{(k-1)} \operatorname{diag}(\mathbf{1}^\top - \boldsymbol{m}_i)
\end{aligned}$$

where $\hat{\boldsymbol{Y}}_{i,\omega:T}^{(k)}$, $\bar{\boldsymbol{Y}}_{i,\omega:T}$, and $\boldsymbol{m}_i$ denote the $i^{\text{th}}$ rows of $\hat{\boldsymbol{Y}}_{\omega:T}^{(k)}$, $\bar{\boldsymbol{Y}}_{\omega:T}$, and $\boldsymbol{m}$, respectively, and $\operatorname{diag}(\mathbf{1}^\top - \boldsymbol{m}_i)$ is the diagonal matrix with the $(j,j)^{\text{th}}$ entry given by the $j^{\text{th}}$ component of the vector $\mathbf{1}^\top - \boldsymbol{m}_i$. Note that $\operatorname{diag}(\mathbf{1}^\top - \boldsymbol{m}_i)$ contains only 1 and 0, this together with $\|\boldsymbol{H}^{(k-1)}\| = s^{(k-1)^2}/(s^{(k-1)^2} + \beta) < 1$ (see Appendix A), where $s^{(k-1)}$ is the largest singular value of $\boldsymbol{R}_{\omega:T}^{(k-1)}$, we obtain

$$\begin{aligned}
\|\boldsymbol{e}_i^{(k)}\| &\le \|\boldsymbol{e}_i^{(k-1)} \boldsymbol{H}^{(k-1)} \operatorname{diag}(\mathbf{1}^\top - \boldsymbol{m}_i)\| + \|\varepsilon_{\bar{\boldsymbol{Y}}}^{(k-1)} \operatorname{diag}(\mathbf{1}^\top - \boldsymbol{m}_i)\| \\
&= \xi \|\boldsymbol{e}_i^{(k-1)} \boldsymbol{H}^{(k-1)}\| + \xi \|\varepsilon_{\bar{\boldsymbol{Y}}}^{(k-1)}\| \le \xi \|\boldsymbol{e}_i^{(k-1)}\| \|\boldsymbol{H}^{(k-1)}\| + \xi \|\varepsilon_{\bar{\boldsymbol{Y}}}^{(k-1)}\|.
\end{aligned}$$

Taking $h = \max\{\|\boldsymbol{H}^{(0)}\|, \ldots, \|\boldsymbol{H}^{(k-1)}\|\} < 1$ and $\bar{\varepsilon} = \max\{\|\varepsilon_{\bar{\boldsymbol{Y}}}^{(0)}\|, \ldots, \|\varepsilon_{\bar{\boldsymbol{Y}}}^{(k-1)}\|\}$ yields

$$\|\boldsymbol{e}_i^{(k)}\| \le \|\boldsymbol{e}_i^{(k-1)}\| \xi h + \xi \bar{\varepsilon} \le \|\boldsymbol{e}_i^{(0)}\| (\xi h)^k + \frac{\xi \bar{\varepsilon}(1 - (\xi h)^k)}{1 - \xi h}.$$

This implies that letting $\delta = \xi \bar{\varepsilon}/(1 - \xi h)$, then $\|\boldsymbol{e}_i^{(k)}\| \le \delta$ as $k \to \infty$.

Furthermore, choosing all the ESN layers as perfect realization ESN yields $\bar{\varepsilon} = 0$, meaning the error sequence $\boldsymbol{e}^{(k)}$ is a monotonically decreasing nonnegative sequence and hence necessarily converges to 0, yielding $\boldsymbol{e}^{(k)} \to 0$. $\qquad\square$

# D  LOWER BOUND ON DL-DRCN LAYER DEPTH FOR A GIVEN IMPUTATION ERROR

**Proposition D.1.** *Given an error tolerance $\varepsilon > 0$, the projection error of the imputation time series satisfies $e^{(k)} < \varepsilon$ whenever $k > \dfrac{\ln\left(\epsilon - \sqrt{q}\delta\right) - \ln\left(\sqrt{q}\|e_j^{(0)}\| - \sqrt{q}\delta\right)}{\ln\left(\xi\right)}$, where $q$ is the dimension of the output time series, and $\delta$ is the error bound defined in Theorem 4.2, and $\xi = \|\boldsymbol{Y}_{1:T}\|/\|\bar{\boldsymbol{Y}}_{1:T}\| = \|\bar{\boldsymbol{Y}}_{1:T} \odot \boldsymbol{M}_{1:T}\|/\|\bar{\boldsymbol{Y}}_{1:T}\|$ with $\boldsymbol{M}_{1:T}$ being the mask of $\boldsymbol{Y}_{1:T}$.*

*Proof.* Applying the bounded condition from Theorem 4.2, the bound of total number of layers $K$ can be derived as follows

$$\begin{aligned}
\|\boldsymbol{e}^{(K)}\|_2 = \sqrt{\|\boldsymbol{e}^{(K)}\|_2^2} &\le \sqrt{\|\boldsymbol{e}^{(K)}\|_F^2} = \sqrt{\sum_{i=1}^q \|\boldsymbol{e}_i^{(K)}\|_2^2} \\
&\le \sqrt{q} \|\boldsymbol{e}_j^{(K)}\|_2 < \sqrt{q} \Big[\|\boldsymbol{e}_j^{(0)}\|(\xi)^k + d_\varepsilon(1 - (\xi)^k)\Big],
\end{aligned}$$

where $\|e_j^{(K)}\| = \max_{i=1}^q \{\|e_i^{(K)}\|\}$, and $h = \max\{\|\boldsymbol{H}^{(0)}\|, \ldots, \|\boldsymbol{H}^{(k-1)}\|\} < 1$. This leads to

$$(\xi)^K < \frac{\epsilon - \sqrt{q} d_\varepsilon}{\sqrt{q}\|e_j^{(0)}\| - \sqrt{q} d_\varepsilon}$$

and therefore

$$K > \frac{\ln\left(\epsilon - \sqrt{q} d_\varepsilon\right) - \ln\left(\sqrt{q}\|e_j^{(0)}\| - \sqrt{q} d_\varepsilon\right)}{\ln\left(\xi\right)}.$$

$\square$

# E  OTHER EXPERIMENTS

| Models | 10% MSE | 10% time (sec.) | 30% MSE | 30% time (sec.) | 50% MSE | 50% time (sec.) | 70% MSE | 70% time (sec.) |
|---|---|---|---|---|---|---|---|---|
| GRU-mean | 6.465±1.581 | 46.548±42.263 | 5.361±0.509 | 55.139±54.471 | 5.039±0.278 | 42.324±31.674 | 4.879±0.147 | 33.069±20.644 |
| GRU-D | 5.611±1.398 | 27.002±7.246 | 4.946±0.444 | 27.598±7.040 | 4.950±0.279 | 25.665±6.147 | 4.919±0.149 | 28.104±6.246 |
| SAITS | 4.988±1.123 | 218.11±15.785 | 4.941±1.076 | 206.61±11.340 | 5.799±2.825 | 204.15±10.303 | 5.277±1.719 | 205.88±9.717 |
| Transformer | 4.925±1.457 | 89.807±27.512 | 4.337±0.562 | 84.324±26.836 | 4.372±0.502 | 79.454±28.388 | 4.558±0.558 | 78.369±35.849 |
| KNN | 5.370±0.892 | 0.610±0.014 | 5.288±0.250 | 0.643±0.008 | 5.893±0.162 | 0.689±0.009 | 7.070±0.249 | 0.801±0.016 |
| MICE | 5.447±1.244 | 7.2±1.9 ($\times 10^{-4}$) | 4.853±0.398 | 7.8±2.9 ($\times 10^{-4}$) | 4.859±0.237 | 8.9±2.7 ($\times 10^{-4}$) | 4.835±0.117 | 8.0±2.8 ($\times 10^{-4}$) |
| CubicSpline | 81.7±65.9 | 12.0±5.1 ($\times 10^{-4}$) | 588.3±701.6 | 12.0±2.6 ($\times 10^{-4}$) | 2.2±1.7 ($\times 10^3$) | 12.0±3.7 ($\times 10^{-4}$) | 3.7±3.1 ($\times 10^3$) | 11.0±1.8 ($\times 10^{-4}$) |
| Linear | 8.138±2.965 | 9.2±2.0 ($\times 10^{-4}$) | 8.277±3.513 | 9.0±1.7 ($\times 10^{-4}$) | 8.431±2.913 | 9.4±3.2 ($\times 10^{-4}$) | 9.710±5.749 | 7.3±1.8 ($\times 10^{-4}$) |
| DLRCN | 0.057±0.020 | 30.138±1.140 | 0.051±0.005 | 29.570±0.694 | 0.054±0.00304 | 30.162±1.398 | 0.063±0.004 | 29.109±0.281 |

Table 4: Time Comparisons of DLDRCN and Other Imputation Methods for Block Missing Scenario: Model performance is evaluated using MSE and total running time (mean ± std) across 40 experiments. It's important to note that in all state-of-the-art models, increasing the dimension of the hidden layers can improve imputation performance, but it also substantially increases computation time.

| Models | Physionet 10% | Physionet 50% | Physionet 90% | ECG 20% | ECG 50% |
|---|---|---|---|---|---|
| Linear | 0.615±0.056 | 1.329±0.099 | 3.502±0.075 | 0.266±0.004 | 0.223±0.028 |
| KNN | 0.662±0.056 | 1.265±0.099 | 3.977±0.085 | 0.232±0.002 | 0.237±0.025 |
| CSDI/SSSD | 0.217±0.001* | 0.301±0.002* | 0.481±0.003* | **0.023±9e-4*** | **0.131±0.003*** |
| DL-DRCN | **0.103±0.001** | **0.128±0.002** | **0.285±0.011** | 0.055±7e-3 | 0.623±0.075 |

Table 5: MAE/RMSE (mean ± std) of imputation results for PhysioNet/ECG datasets.

# F  EVALUATION METRICS

For a given ground truth data matrix $\boldsymbol{Y} \in \mathbb{R}^{q \times T}$ and the reconstruct outcome $\hat{\boldsymbol{Y}} \in \mathbb{R}^{q \times T}$ from different models, we evaluate the imputation performances using the mean square error (MSE) and mean absolute error (MAE), given by

$$\text{MSE} := \|\hat{\boldsymbol{Y}} - \boldsymbol{Y}\|_F^2 = \frac{1}{qT} \sum_{i=1}^q \sum_{j=1}^T (\hat{y}_{i,j} - y_{i,j})^2;$$

$$\text{RMSE} := \|\hat{\boldsymbol{Y}} - \boldsymbol{Y}\|_F = \sqrt{\frac{1}{qT} \sum_{i=1}^q \sum_{j=1}^T (\hat{y}_{i,j} - y_{i,j})^2};$$

$$\text{MAE} := \frac{1}{qT} \sum_{i=1}^q \sum_{j=1}^T |\hat{y}_{i,j} - y_{i,j}|.$$

# G  TOTAL FLOP COUNTS OF DL-DRCN

The total *floating point operations (flops)* counts of our DL-DRCN algorithm at each step are summarized in the following table, where $N, p, q, d$ are the dimensions of $\boldsymbol{R}$ (reservoir matrix), $\boldsymbol{U}$

(input), $\boldsymbol{Y}$ (output), $\boldsymbol{X}$ (dataset), respectively. $T$ is the number of total timesteps and $S$ is the total number of non-zero elements in the sparse matrix $\boldsymbol{A}$.

| Step | flop count |
|------|-----------|
| Compute the evolution of reservoir state | $(\underbrace{N}_{(1-\alpha)\boldsymbol{r}_{t-1}} + \underbrace{N}_{[\cdot]+[\cdot]} + \underbrace{N}_{\alpha^{(k)}\tilde{\boldsymbol{r}}_t^{(k)}} + \underbrace{2S}_{\boldsymbol{A}^{(k)}\boldsymbol{r}_{t-1}^{(k)}} + \underbrace{2Np}_{\boldsymbol{B}^{(k)}\boldsymbol{u}_t}) * T$ |
| Compute the weight matrix | $\underbrace{2TNq}_{\boldsymbol{Y}*[\cdot]} + \underbrace{2N^2T}_{\boldsymbol{R}*[\cdot]} + \underbrace{2N^2T}_{\boldsymbol{R}\ \boldsymbol{R}^\top} + \underbrace{2N}_{+\beta\boldsymbol{I}} + \underbrace{2N^3}_{(\cdot)^{-1}}$ |
| Compute update | $\underbrace{qT}_{\boldsymbol{Y}\odot\boldsymbol{m}} + \underbrace{qT}_{[\cdot]+[\cdot]} + \underbrace{qT}_{[\cdot]\odot(\boldsymbol{1}-\boldsymbol{m})}$ |
| Total *flop* counts | $4N^2T + 2N^3 + 2dNT + 2ST + 3NT + 3qT + 2N$ |

Table 6: Hyperparameters of DL-DRCN

## H DL-DRCN HYPERPARAMETERS FOR EXPERIMENTS

As mentioned in Section 4.1, the predetermined hyperparameters in RCN models include: $\sigma^{(k)}$ denotes the element-wise activation functions, which we choose the nonlinear hyperbolic tangent function (tanh) for each iteration in our experiment; $\boldsymbol{A}^{(k)} \in \mathbb{R}^{N^{(k)} \times N^{(k)}}$ denotes the weighted adjacency matrix of the reservoir layer, which is obtained by first randomly generate a sparse matrix $\tilde{\boldsymbol{A}}^{(k)}$, then the adjacency matrix $\boldsymbol{A}^{(k)}$ is derived as $\boldsymbol{A}^{(k)} = \tilde{\boldsymbol{A}}^{(k)}/|s^{(k)}|L^{(k)}$ with $s^{(k)}$ being the largest eigenvalue of $\tilde{\boldsymbol{A}}^{(k)}$ and $L^{(k)}$ being the Lipschitz constant of the activation function $\sigma^{(k)}$. The purpose of this process is to guarantee the ESP for each ESN layer, or equivalently $\|\boldsymbol{A}^{(k)}\| < 1/L^{(k)}$ for each layer. Note that the Lipschitz constant of the activation function is equal to 1, i.e., $L^{(k)} = 1$, due to the choice of the tanh function; $\boldsymbol{B}^{(k)} \in \mathbb{R}^{N^{(k)} \times p}$ delineates the input weight matrix, in which each element in $\boldsymbol{B}^{(k)}$ is chosen randomly from a uniform distribution; $\alpha^{(k)} \in (0,1]$ denotes the leakage rate, which is chosen to be close to 1. Table 7 summarizes all the hyperparameters used in our DL-DRCN model. Additionally, the readout map in each ESN layer is chosen to be a linear map, resulting in a simple linear relation between the output state and the reservoir state, of the following form $\boldsymbol{Y} = \boldsymbol{C}\boldsymbol{R}$. As described in section 3.2, finding the optimal weight matrix is equivalent to solving a least square problem, where a regularization term is further considered in this case to prevent the overfitting problem.

|   | Hyperparameters | | Values |
|---|---|---|---|
| | $\boldsymbol{A}$ | reservoir adjacency matrix | $\rho_A = \|\boldsymbol{A}\| = 1$ |
| | $\boldsymbol{B}$ | input weighted matrix | $b_{i,j} \in [-1,1]$ |
| $\Lambda$ | $\alpha$ | leakage rate | $\alpha = 0.8$ |
| | $\sigma$ | activation function | $\sigma = \tanh$ |
| | $\beta$ | regularization parameter | $\beta = 10^{-8}$ |

Table 7: Hyperparameters of DL-DRCN

We fill in the missing values with linear interpolation method as initial values for ECG and Physionet experiments. we chose The dimensions of ESN layers we chose for each experiment are listed in following Table 8.

| Experiments | $N^{(k)}$ | Total layer numbers |
|---|---|---|
| Rössler System | $1000, 975, 950, \ldots, 500$ | 21 |
| Gesture | $1600, 1570, 1540, \ldots, 1000$ | 21 |
| ECG | $2000, 2000, 2000, \ldots, 2000$ | 20 |
| Physionet | $200, 200, 200, \ldots, 200$ | 20 |

Table 8: Hyperparameters of DL-DRCN

## I  DATA DESCRIPTION AND PREPROCESSING

### I.1  SYNTHETIC DYNAMICAL SYSTEM

In this synthetic data example, we generated a multivariate time series using the Rössler system, whose dynamics are given by,

$$\dot{x}_t = -y_t - z_t, \quad \dot{y}_t = x_t + ay_t, \quad \dot{z}_t = b + z_t(x_t - c), \quad (6)$$

where $(x_t, y_t, z_t)^\top \in \mathbb{R}^3$ is the state variable at time $t$ and $\{a, b, c\}$ are constant parameters. Here, $(\cdot)^\top$ denotes the transpose of the vector. In particular, we choose $a = 0.5$, $b = 2.0$, and $c = 4.0$ and solve this system of differential equations by using a 4th-order Runge-Kutta based ODE solver (*ode45*) in Matlab. The system is solved from the initial condition $(x_0, y_0, z_0)^\top = (0, 0, 1)^\top$ over the time interval $[0, 320]$ with the sampling rate 0.04, yielding a multivariate time series with the spatial and temporal dimensions 3 and 8000, respectively.

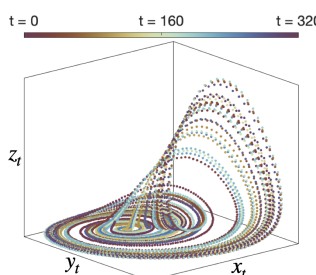

Figure 5: State trajectories $x_t$, $y_t$, $z_t$ of the Rössler system.

### I.2  PHYSIONET DATASET

In this experiment, we collected the health-care clinical dataset from *PhysioNet Challenge 2012*, which is a publicly available dataset containing multivariate clinical time series extracted from the Multiparameter Intelligent Monitoring in Intensive Care II (MIMIC II) database. In total 12000 patient ICU records were selected randomly from the data pool and were divided equally into three groups, training set A, testing set B, and testing set C, with each containing 4000 patients' records. Since testing set C is blinded and unavailable to the public, we only collected the remaining 8000 records for this study. Each patient record contains up to 41 clinical variables which were measured irregularly from the first 48 hours after the patient's admission to ICU. Since not all variables were recorded

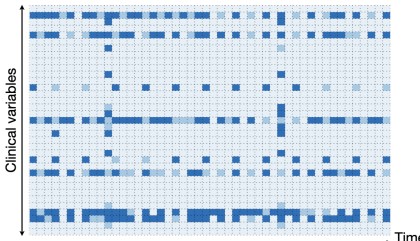

Figure 6: An example mask showing 30% data removal for imputation on the Physionet dataset, with light blue highlighting the removed data points.

for each patient and each variable was measured at different time points, we followed the preprocessing steps in (Tashiro et al., 2021; Che et al., 2018) by selecting 35 out of 41 variables as features and rounded up the time stamps to 1 hour, resulting in a multivariate time series of 35 features and 48 points per feature.

### I.3  GESTURE DATASET

The gesture phase segmentation dataset is collected from (Madeo et al., 2014), where the dataset comprises features extracted from 7 video recordings with people gesticulating. There are in total 14 (*.csv*) files included, with 1 raw file and 1 processed file for each video recording. The files are categorized by three test users (A,B,C) and the stories (1,2,3) each subject is being asked to read and present in front of the sensors. Specifically, these files include A1, A2, A3, B1, B3, C1 and C3, with each containing around 1000-1800 frames. We study the processed video dataset in A1 (*a1va3.csv*), which contains 32 features and 1743 datapoints per feature.