# OpenReview forum: "Layer-Varying Deep Reservoir Computing Architecture"
_ICLR.cc/2025/Conference — Submitted to ICLR 2025_

### Official Review · Reviewer_CHYy · 2024-10-24

**Soundness:** 2
**Presentation:** 3
**Contribution:** 2
**Rating:** 5
**Confidence:** 4

**Summary:**

The paper proposes a framework to perform imputation of multivariate time series with Reservoir computing. In particular a stack of Reservoirs is used to learn increasingly more complex (and arguably expressive) representations of the underlying dynamical systems observed through the time series. Those representations are used to impute the missing values.

**Strengths:**

- The approach is rather straightforward and easy to implement.
- The authors provide a theoretical analysis of the structure of the Reservoir model necessary to achieve a desired imputation error.

**Weaknesses:**

1. Contribution. The idea of using Reservoir computing to perform imputation of missing data is not novel. See for example my comment about the missing references. The main contribution seems to be the theoretical derivations, but I have some issues with those (see next point).

2. To my understanding, the theoretical results rely on the assumption that the Reservoir, if sufficiently large, contains the ground truth $\bar{Y}$, i.e., the complete time series with also the true values that are missing. If that is the case, such information can be easily retrieved by a linear readout. Such a ground truth should emerge from the Reservoir dynamics, which is driven by the input U. However, I am skeptical that this would happen in practice. What, for example, if U is just a constant input? How the Reservoir dynamics could contain any possible time series $\bar{Y}$, e.g., a time series representing a non-stationary process, such as the stock prices of a given company, the number of sales of a given product, the number of cars going through a cross-road, and so on? The paper makes no assumptions about relationships between U and Y, making it difficult for me to believe that *any* U could produce *any* desired $\bar{Y}$. In addition, due to the ESP property, the Reservoir cannot learn a whole set of time series, e.g., those that are not stationary, that are very chaotic, and so on.

3. There are some strong assumptions (see more in the questions).

4. There are some imprecisions. For example, the ESP depends also on the leakage $\alpha$.

5. Some minor issues with the notation. For example, the \bar is used both in line 216 and to express the ground truth. Also, in 229 and 231 "w" is used instead of omega. In 307 should be $\mathcal{K}$ rather than $k$?

6. Some missing references. About the claim in lines 100-101, some recent papers such as [1] should be referenced and contextualized with the current paper. About time series imputation with Reservoir computing some relevant literature should be cited and compared with the current paper, see e.g., [2, 3]. Literature about the ESP in deep Reservoir computing should be cited, see e.g., [4].

[1] Marisca, Ivan, Cesare Alippi, and Filippo Maria Bianchi. "Graph-based forecasting with missing data through spatiotemporal downsampling." International Conference on Machine Learning, 2024.

[2] Huang, Fangwan, et al. "Estimating missing data for sparsely sensed time series with exogenous variables using bidirectional-feedback echo state networks." CCF Transactions on Pervasive Computing and Interaction 5.1, 2023.

[3] Wang, Qiang, et al. "Time series prediction with incomplete dataset based on deep bidirectional echo state network." IEEE Access, 2019.

[4] Gallicchio, Claudio, and Alessio Micheli. "Echo state property of deep reservoir computing networks." Cognitive Computation, 2017.

**Questions:**

- Can you handle the univariate time series case, i.e., when there is no $\mathbf{U}$?
- There are some strong assumptions, such as not all values can be missing from every covariate of the time series at once. I believe this is somewhat a limitation because in real-world scenarios that is something that actually happens. Imagine if the variables of the time series are different measurements collected from the same sensor that goes offline and stops recording all of them. In addition, there is an assumption that missing values are not allowed at the beginning of the time series, i.e., when the Reservoir is still going through a transient phase. What can be done if that happens?

---

> ### Author Response · Authors · 2024-11-24
>
> # Response to Weaknesses
>
> 1. The main contribution lies in the development of the deep reservoir computing architecture composed of a cascade of ESN layers with diminishing reservoir sizes, which is beyond directly adopting classical reservoir computing for time series imputation.
>
> 2. In the context of imputation, the ground truth $\bar Y$ is unknown, in which case, regardless of $\bar Y$ lying in the reservoir space or not, it is necessary to learn it iteratively. In addition, because the reservoir space is spanned by the reservoir states generated by the reservoir dynamics, $\bar Y$ indeed emerges from the dynamics. Moreover, as defined in Lines 118 to 126 in Section 3.1, $U$ and $Y$ are composed of different components of the same multivariate time series, and hence necessarily correlated. The proposed DL-DRCN indeed fully explores and utilizes their correlation to infer the missing values in $Y$. On the other hand, we would like to politely point out that ESNs have been proven to perform well in modeling non-stationary, particularly chaotic, systems [1-2]. The synthetic datasets used in our work were generated by a chaotic system, the R\"{o}ssler system.
>
> [1] Mantas Lukosevicius, Herbert Jaeger, ``Reservoir computing approaches to recurrent neural network training.''
>
> [2] H. Jaeger and H. Haas, “Harnessing nonlinearity: Predicting chaotic systems and saving energy in wireless communication.”
>
> 3. Please refer to Response to Questions \#2.
>
> 4. As shown in Lemma B.2, ESP holds regardless of the leakage $0<\alpha\leq 1$, and only the Lipchitz constant of the reservoir map depends on $\alpha$.
>
> 5. Corrected.
>
> 6. The references pointed out by the reviewer has been cited in the revised manuscript.
>
>
> # Response to Questions:
>
> 1. Our approach can handle univariate time series as well. In such a case, say $X=[x_1,\dots,x_T]$ with $x_t\in\mathbb{R}$ for all $t=1,\dots,T$, we utilize the delay embedding to construct a multivariate time series
> $$
> \tilde X=\big[\tilde X_1 , \tilde X_2 , \cdots , \tilde X_\tau\big]=\left[\begin{array}{cccc} x_1 & x_2 & \cdots & x_\tau \\\\ x_2 & x_3 & \cdots & x_{\tau+1} \\\\ \vdots & \vdots & \cdots & \vdots \\\\ x_{k} & x_{k+1} & \cdots & x_{k+\tau} \end{array}\right]
> $$
> with $k+\tau=T$, then DL-DRCN can be applied to $\tilde X$.
>
> 2. These two cases mentioned by the reviewer are also pointed out in the Limitation and future works section of the paper. However, a possible approach to imputing the missing values located at the beginning of a time series is to learn the inverse evolution, meaning $y_{t-1}=f(y_t,u_{t-1})$ in the notation used in Section 3.2. This can be realized by rearranging the time series $X=[X_1 , X_2 , \cdots , X_T]$ backward in time as $\tilde X = [X_T , X_{T-1} , \cdots , X_1]$, and then apply DL-DRCN to $\tilde X$.

---

> > ### Comment · Reviewer_CHYy · 2024-11-25
> > **My concerns remain after the rebuttal**
> >
> > I am not particularly satisfied by the author's rebuttal, which felt rushed. The authors did not do a good job in clarifying my doubts and concerns, which persist.
> >
> > After having read the rebuttal that the authors provided to me and to the other reviewers, the novelty of the proposed architecture, seems to be a small variant of architectures that already exist.
> >
> > About the assumptions, lines 118 and 126 do not mention anything about the correlation between the covariates in the multivariate time series. If that were the case, I would not have come up with my doubt that it is not possible to infer $\bar{Y}$ if the other covariates represent something completely different.
> > The claim that covariates belonging to the same time series are always correlated is not necessarily true.
> > Also, the paper does not mention such a correlation, which, in my opinion should be a central point of the analysis. Clearly, if some values are missing from a covariate i, which is very strongly correlated with another covariate j that is not missing, it is obvious that the missing values can be recovered to some extent. On the other hand, the cornerstone of the theoretical derivation is that the Reservoir somehow already contains the time series that should be imputed, which is an assumption that seems too strong. This can maybe relate to the universal approximation theorem since the authors assume their theoretical results to hold for $k \rightarrow \infty$, but is not clear to me and the authors did not make it clearer in their rebuttal.
> >
> > Also, an ESN might be able to predict a time series generated by a chaotic system such as Rossler (which, to be fair, is a rather easy task that can be solved by regression on the time-delay embedding). However, it fails in predicting non-stationary signals such as a linear or a quadratic trend. Therefore, I am still convinced that the Reservoir could not contain, at least in practice, a $\bar{Y}$ equal to such a trend and, thus, I remain unconvinced about the theoretical part of this paper.
> >
> > About the ESP and alpha, the sufficient condition for ESP surely depends also on alpha because it affects the Lipschitz coefficient which should be <1. In fact, alpha affects the eigenvalues of the Jacobian.

---

### Official Review · Reviewer_A3V4 · 2024-10-27

**Soundness:** 3
**Presentation:** 2
**Contribution:** 3
**Rating:** 5
**Confidence:** 4

**Summary:**

The authors model a multivariate time series as a trajectory within a dynamical system and develop a deep reservoir computing architecture composed of multiple echo state network (ESN) layers with progressively smaller reservoir sizes. They introduce a layer-by-layer training method that leverages the echo state property of ESNs, which aids both in convergence and efficiency. Through experiments with synthetic and real-world datasets, they demonstrate the effectiveness of this model in accurately imputing missing or corrupted data.

**Strengths:**

This article is clearly presented, with adequate theoretical and experimental support, effectively demonstrating the proposed DL-DRCN method. This work plays an important role in the task of interpolation in multivariate time series.

**Weaknesses:**

This work primarily conducts research based on the reservoir computing, however, the introduction section lacks discussions on the latest research concerning the reservoir computing method, such as references [1] and [2].

Furthermore, the author lacks significant ablation experiments. specifically, the practical difference between traditional reservoir computing and multi-layer reservoir computing.  I even think that merely utilizing a reservoir computing, repeatedly employing Equation 3 for interpolation, could also achieve relatively good results.




[1] Gauthier D J, Bollt E, Griffith A, et al. Next generation reservoir computing[J]. Nature communications, 2021, 12(1): 1-8.

[2] Li X, Zhu Q, Zhao C, et al. Higher-order Granger reservoir computing: simultaneously achieving scalable complex structures inference and accurate dynamics prediction[J]. Nature Communications, 2024, 15(1): 2506.

**Questions:**

1. Section 3.1's first paragraph has some unclear statements regarding $X, Y, U, M$.

2. Proposition 3.1 does not seem to be a new theoretical result proposed by the author. If not, it needs to be cited, and including it in the main text is inappropriate.

3. Section 4.2, the first line contains a typographical error.

4. What concerns me is, what are the advantages of the multi-layer reservoir computing approach proposed by the author compared to the single-layer one? To my knowledge, the single-layer reservoir computing method can also perform well in the task of reconstructing the Rossler system, and according to Takens' Embedding Theorem, partial observations can still learn the dynamics of the system.

---

> ### Author Response · Authors · 2024-11-24
>
> # Response to Weaknesses:
>
> We have cited the two references mentioned by the reviewer in the revised manuscript.
>
> Traditional reservoir computing is essentially one iteration (layer) of the proposed DL-DRCN. Therefore, the imputation performance of traditional reservoir computing has already been included in Figure 3 as the case of the total number of iterations equal to 1. In the revised manuscript, we have also reported these results in Table 1. Moreover, the proposed DL-DRCN exactly involves employing Equation 3 iteratively, for which a reservoir computing network is necessary to generate the regressors for interpolation.
>
>
> # Response to Questions:
>
> 1. As a clarification, $X$ is the entire multivariate time series, $Y$ consists of the components of $X$ with missing values, $U$ is the components of $X$ complementary to $Y$, and $M$ is the mask for the missing values.
>
> 2. There's a wide variety of results regarding ESP for a vanilla ESN, for instance, see [2-3]. However, in terms of deep reservoir computing architecture, the ESP condition is different from that proposed in [1]. As a result, we provide a sufficient condition of ESP for our framework.
>
> [1] Claudio Gallicchio, Alessio Micheli, ``Echo State Property of Deep Reservoir Computing Networks.''
>
> [2] Michael Buehner, and Peter Young, ``A Tighter Bound for the Echo State Property.''
>
> [3] Izzet B. Yildiz, Herbert Jaeger, and Stefan J. Kiebel, ``Re-Visiting the Echo State Property.''
>
> 3. Corrected.
>
> 4. The main advantage is to improve the imputation performance as shown in Figure 3, from which we observe a significant decrease of the MSE with respect to the number of iterations (layers).

---

> > ### Comment · Reviewer_A3V4 · 2024-11-25
> >
> > Thank you for your feedback. I apologize as I perceive that the article's contributions and innovations require further elucidation, particularly concerning the advantages of a multi-layered RC over a single-layered RC. If the initial layers' RC serve merely to interpolate time series for the purpose of data augmentation, then the necessity of this framework appears insufficient to warrant a positive score from my perspective. Consequently, I will maintain the current score.

---

### Official Review · Reviewer_PnqC · 2024-11-03

**Soundness:** 2
**Presentation:** 3
**Contribution:** 3
**Rating:** 5
**Confidence:** 4

**Summary:**

The authors study data-imputation problem in multivariate time series, and consider random- and block-missing structures. To approach this problem, the authors propose a ESN-based approach which refines the imputations at each iteration by projecting the representations of the observed covariates onto the imputed sequence for each layer. The representations at each layer are computed using a reservoir with decreasing number of neurons at each layer.
The authors display this approach's benefits on a synthetic dynamical system dataset, and 3 real-world datasets.

**Strengths:**

1. The paper is well writen and easy to follow.
2. The method appears appears to be well-motivated, the burdens of the iterative process devised by the authors are offset by the lightweight computational costs of ESNs. Overall, the method appears elegant.
3. Results are positive and underscore the benefits of the proposed architecture.

**Weaknesses:**

1. Comparison against vanilla ESN baselines is lacking:
   - ESNs are known to perform well on chaotic time series. Comparisons should include a vanilla ESN with basic imputation methods.
   - Additionally, it would be informative to show qualitative results for a vanilla ESN, similar to Fig. 2.
2. Insufficient comparison with related methods on more realistic datasets:
   -  Comparisong in Table 3: only cases with 10% missing values are shown; the original paper also includes 50% and 90% missingness.
3. The claim on convergence of the algorithm appears to be misleading.
   - I believe this to be a typo. For instance, you claim "The sequence of imputed time series resulting from this algorithm is guaranteed to converge to the ground truth time series...". I believe the authors mean to claim the method converges to the ground truth known target values $Y_{\omega:T}$, not $Y_{1:T}$. Claiming that the imputations can recover the unknown ground truth sequence would be puzzling.

**Questions:**

1. Why did you use ESNs specifically for this application over other recurrent NNs, e.g. GRUs? Was efficiency the main factor here?
2. In the "Dynamical systems-theoretic approach to time series analytics" section, why is $u_t$ defined as $u_t \in \mathbb{R}^{q \times p}$? Isn’t $u_t$ $p$-dimensional?
   - Additionally, $y_t$ is assumed to depend only on $y_{t-1}$ and $u_t$, but not $u_{t-1}$. Are there no carry-over effects? While $y_{t-1}$ indirectly depends on $u_{t-1}$, is this sufficient?
3. The method’s efficacy is challenging to understand intuitively. How does independent hidden state generation $r_t^k$ at each layer improve predictions, given it doesn’t rely on prior target predictions? It’s also puzzling that a 1-layer DL-DRCN (up to additional hyperparameters equivalent to a vanilla ESN on known target states) underperforms.

Small details:
1. The authors write: "The main complexity of our algorithm comes from the matrix inverse operation, yielding an $O(KN^2T)$ complexity with $N$ as the number of neurons in the first ESN layer and $K$ as the total number of ESN layers," but matrix inversion typically requires $O(N^3)$ operations. This is expressed correctly in Table 4 of the Appendix.
2. The distinction between $Y_{1:T}$ and $Y_{\omega:T}$ in Fig. 1 is confusing, as the caption refers to the latter while the figure uses the former notation. Could this notation be unified for clarity in the figure?


Overall I'd be open to increase my score by at least 1 point if the weaknesses and the questions are addressed.

---

> ### Author Response · Authors · 2024-11-24
>
> # Weaknesses:
>
> 1. Imputation performance of vanilla ESNs have been included in Figure 3, which are essentially the MSE of one iteration of the proposed algorithm. We have now summarized these results in Table 1 as well.
>
> 2. We have included the 50\% and 90\% missing cases in Appendix Section E.
>
> 3. As mentioned on Line 231 of the manuscript (Line 236 of the revised manuscript), $\omega$ is chosen in the way that $Y_{1:\omega-1}$ does not contain missing value. Therefore, convergence of $Y_{\omega:T}$ is equivalence to that of the entire time series $Y_{1:T}$.
>
> # Questions:
>
> 1. In addition to the efficiency advantages ESNs provide, our framework leverages the linear form of ESN's readout map to establish the projection mechanism. In particular, one key contribution of our work is the introduction of an ``iterative training process," wherein we propose an update equation to iteratively train and improve the model's performance. This approach is uniquely enabled by the formulation presented in our work.
>
>
> 2. Yes, $u_t$ is indeed a $p$-dimensional vector, and this typo has been corrected. In addition, it is sufficient to consider the dynamical system model $y_t=f(y_{t-1},u_t)$, where the system evolution $f$ depends on the previous system state $y_{t-1}$ and current system input $u_t$ only, because systems with ``carry-over effects'' can also be transformed to this form. Specifically, given a system $y_t=g(y_{t-1},y_{t-2},\cdots,y_{t-\tau},u_t,u_{t-1},\dots, u_{t-\tau+1})$ with $y_s\in\mathbb{R}^q$ and $u_s\in\mathbb{R}^p$ for all $s$, we define $Y_{t-1}=[y_{t-\tau}^\intercal,\dots,y_{t-1}^\intercal]^\intercal\in\mathbb{R}^{\tau q}$, $U_t=[u_{t-\tau+1}^\intercal,\dots,u_{t}^\intercal]^\intercal\in\mathbb{R}^{\tau p}$, and $f(Y_{t-1},U_t)=[y_{t-\tau+1}^\intercal,\dots,y_{t-1}^\intercal,g(y_{t-1},y_{t-2},\cdots,y_{t-\tau},u_t,u_{t-1},\dots, u_{t-\tau+1})^\intercal]^\intercal\in\mathbb{R}^{\tau q}$, then the system can be equivalently represented in the form $Y_t=f(Y_{t-1},U_t)$.
>
> 3. The reservoir state $r_t^k$ of the $k^{\rm th}$ ESN layer does depend on that $r^{k-1}_t$ of the previous ESN layer since the input of the $k^{\rm th}$ layer is the time series imputed by the $(k-1)^{\rm th}$ layer. Under the assumption that the reservoir space of each ESN is large enough to contain the ground truth, it guarantees that the $k^{\rm th}$ layer projects the time series to be imputed to a smaller space containing the ground truth than the $(k-1)^{\rm th}$ layer does. Therefore, the imputation performance is improved layer-by-layer, which in turn explains the reason for underperformence of 1-layer DL-DRCN.
>
> Small details:
>
> 1. The complexity of matrix inverse (inverse of a $N\times N$ matrix) requires $O(N^3)$ operations; however, in our case, we are calculating the pseudoinverse of a $N\times T$ matrix with $N<T$, yielding a complexity of $O(N^2T)$.
>
> 2. The figure has been corrected.

---

> ### Comment · Reviewer_PnqC · 2024-11-26
>
> I thank the authors for their response to my review. Most of my concerns have been addressed satisfactorily. However, I would like to raise two potential misunderstandings for further clarification:
>
> 1. Regarding the statement, *"If $\omega$ is chosen in the way that $Y_{1:\omega-1}$ does not contain missing values, then the convergence of $Y_{\omega:T}$ is equivalent to that of the entire time series $Y_{1:T}$"*: Does this imply that convergence is, naturally, guaranteed only for the observed parts of the sequence?
> 2. Could you point out the specific lines in Algorithm 1 where the dependence of predicted inputs on the preceding layer is explicitly utilized in the dynamics $r$?
>
> Additionally, I have some suggestions for the authors to consider, although these do not affect my overall score:
>
> 1. Several reviewers have noted the similarity of the authors’ approach to deepESN architectures. While I personally do not find significant conceptual overlap, conducting an experimental comparison with deepESN could help clarify distinctions between the works and strengthen the contribution.
> 2. I also agree with *Reviewer 3KyE* that the notation could be simplified. A short index summarizing the key notation introduced in the paper, along with their conceptual roles, would greatly enhance the paper's clarity.

---

### Official Review · Reviewer_XLK9 · 2024-11-03

**Soundness:** 2
**Presentation:** 1
**Contribution:** 1
**Rating:** 3
**Confidence:** 4

**Summary:**

The work discusses an application of deep echo state networks (ESN( to timeseries imputation. Missing measurements are incrementally imputed in the multivariate timeseries through predictions generated by the readouts attached to the different layers of the ESNs. Experiments are provided to validate the efficacy of the methodology against related timeseries imputation methodologies and 4 datasets.

**Strengths:**

* Imputation in multivariate timeseries, especially when considering missing whole blocks of observations, is an interesting topic for the community working on spatio-temporal data and may achieve good impact.

 * The proposed approach is well rooted into the literature of efficient timeseries processing via reservoir computing methods: design choices are solid in this sense.

**Weaknesses:**

* Originality of the approach is minor. The proposed methodology is essentially using a standard deepRC/deepESN architecture with read-outs for each layer (an architectural pattern already explored by Gallicchio et al, 2017 and Gallicchio et al, 2018, among the others). The key catch of the approach is that a read out at layer k is trained to predict masked/missing dimensions of the multi-variate timeseries using observable dimensions and a self-generated target from the previous layer. Section 2 (last line) contains a claim of novelty against previous deepRC approaches (being "fundamentally different") but no discussion is provided to substantiate what is such substantial difference. Architecturally the approach seems very much the same of earlier works, with the addition of a self-supervised incremental training and the application to timeseries imputation.

 * Clarity and quality of the presentation are also limited. The manuscript is particularly unclear about the separation between literature results and own novel contributions. An example has been given in the comment above. Another example is the attribution of results on reservoir contractivity and the dynamical systems interpretation of timeseries and reservoirs. These are widely known results from literature which are not appropriatly acknowledged, e.g. by providing bibliographic references in Section 3. Also, the paper mixes reservoir computing with echo state networks without making explicit the difference between the two (in fact ESNs are used without being introduced and a sort of synonimity between RC and ESN is hinted at in the paper, which is wrong): this is confusing for a novice of the RC field. Low presentation quality is also evident from wording errors which could easily spotted with any syntax checking tool (not even grammar checkers): e.g. "wigth matrix" in page 4, "groudtruth" in page 6, etc.

 * The soundness of the theoretical results is not entirely convincing. The error bounds in page 6 (Theorems and corollary) seem to be demonstrated for the $Y_w:T$ part of the timeseries, which is convincing as this is the target of the readout training on those portions of the timeseries which have no missing values (at least this is what stated in page 5). Now, what it is unclear to me is how this result can be generalized for portions of the timeseries outside these $w:T$ intervals (which are those where the observations are actually missing). I am not doubting how these can be inferred using the trained read-outs. It is not clear to me how the error bounds apply to those missing portions of the timeseries.

 * The empirical analysis is limited in scope and depth of the discussion. Scope of the comparison should be enlarged to include those works approaching time-series imputation from a spatio-temporal perspective (e.g. https://www.ijcai.org/proceedings/2024/0445.pdf). The results provided are not analyzed in depth: for instance in Table 2 top, the confidence intervals seem to be very much overlapped with those of GRU-D and no discussion on the significance of these results is provided.

 * The work does not make any attempt at using the insight provided by earlier works on deep reservoirs (i.e. Gallicchio et al, 2017) to explain the effect of multiple reservoirs on the timeseries. E.g. it is know from earlier literature that layers in a deep reservoir architecture tend to produce an incremental filtering effect in which each layer focuses on filtering out selective frequencies out of the input timeseries. The impact of such a process on the imputation application should be analysed more in depth.

**Questions:**

1) Can you please elaborate on the statistical significance of the performance differences in Table 2?

2) Given the strong inductive bias of reservoir computing methods (due to the restrictions imposed by the randomization of weights), it is largely surprising that the hyperparameters do not vary across datasets (aside from reservoir size, which seems to be the only one in model selection). Can the authors elaborate more on this? For instance, the leaky parameter can be expected to be different between datasets as its choice typically depend on the different “speed” of the signals involved. Similarly for the spectral radius and, to a minor extent, to the effect of sparsity.

3) Can you please clarify how the theoretical results generalize to parts of the sequences that need imputation and where this can be tracked in the demonstrations provided in the appendices?

---

> ### Author Response · Authors · 2024-11-24
>
> # Weaknesses:
>
> 1. The proposed deep reservoir computing (RC) architecture, composed with a cascade of ESN layers with diminishing reservoir sizes, is novel and fundamentally different from the existing deep RC structure, notably the one presented in Gallicchio et al, 2017 as mentioned by the reviewer, from both conceptual and technical perspectives. Specifically, the one in Gallicchio et al, 2017 has the conceptual interpretation in terms of the multiple time-scale representation; however, ours, following from the use of the read-out for each layer, is an iterative projection of the input time series onto a descending sequence of vector spaces containing the ground truth. Technically, we carry out a layer-by-layer "regression style" training scheme in accordance with the iterative projection idea, which is also different from the widely used backward propagation method of training neural networks.
> 2. We politely disagree with the reviewer's comment on the separation between literature review and the claim of the contributions, which are entirely in different sections, Sections 1 and 2, respectively. Moreover, we did cite quite a few references to acknowledge existing relevant works. On the other hand, we are proposing a deep RC architecture with cascade of ESN layers. Hence, we are not sure where the confusion is regarding RC and ESN. We will be happy to address the reviewer's specific concerns.
> 3. We would like to correct the reviewer that $Y_{\omega:T}$ is indeed the part containing missing values. This directly follows from the choice of $\omega$ such that $Y_{1:(\omega-1)}$ does not contain missing values on Line 231 of the original manuscript.
> 4. We appreciate the reviewer’s feedback on expanding the scope and depth of our empirical analysis. Regarding the comparison to spatio-temporal approaches, while our study primarily focuses on general time-series imputation frameworks, we acknowledge the importance of considering these specialized methods and will incorporate them into future analyses to provide a more comprehensive evaluation.
> 5. We would like to emphasize there are some fundamental differences between Gallicchio's work and ours. One fundamental difference between these two works lies in the structural design differences. In Gallicchio's deepESN, the emphasis is mainly placed on the ``state transition function'' (referred to as the reservoir map in our terminology), where the input to each ESN layer is the reservoir state from the preceding layer. In contrast, our framework uses the same input time series across all ESN layers. A key contribution of our work is the introduction of an "iterative training process," wherein we propose an update equation to iteratively train and improve the model's performance. This approach is uniquely enabled by the formulation presented in our work. Additionally, our framework simplifies the analysis, e.g., the echo state property, of deep reservoir computing architectures compared to the approach used in deepESN.
>
> #Questions:
> 1. In both Tables 1 and 2, each experiment was repeated 40 times with each time using different masks to create missingness. We then reported results including the mean and standard deviation of the mean squared error across these repetitions.
> 2. The layer-by-layer decrease in reservoir size is a novelty of the proposed DL-DRCN, which is also one of the main contributions of this work. Therefore, the reservoir size is not treated as a hyperparameter here. On the other hand, the hyperparameters were chosen based on the ablation studies carried out in Section 5.1. Taking the leaky parameter pointed out by the reviewer as an example, the MSE remains small for a wide range of this hyperparameter, demonstrating the robustness of the proposed DL-DRCN to the leaky rate.
> 3. The theoretical result proposed in Theorem 4.2 is described in terms of the error between the ground truth time series and the reconstructed time series; and this error bound can be easily generalized to the sequences needed to be imputed by multiplying the mask term. The whole derivation remains consistent as provided in Theorem 4.2.

---

> > ### Comment · Reviewer_XLK9 · 2024-11-25
> > **On rebuttal**
> >
> > I ackknowledge the rebuttal and thank the reviewers from their response.
> >
> > First one comment, as the authors asks for more clarity about my comment on RC Vs ESN. My point is that in the paper, as well as in the Authors response, there is confusion between RC and ESN. Sticking to the rebuttal, it is stated that the approach is a DeepRC with ESN layers. ESN are not layers. They are one specific implementation of a model from the RC family. Specifically, an ESN is a model with randomized inputs and reservoir layer, initialized under contractivity conditions and with a linear read-out. It is not a layer.
> >
> > Coming down to the gist of the rebuttal: the comments from the Authors are not shedding further light on the novelty of the model. If the novelty of the approach is on the deep reservoir with reservoirs layers incrementally halved in size and a specific input being fed to them, I do not think such a methodological contribution to be sufficient for ICLR. Especially as the work then is not compensating the reduced methodological novelty with an evidence of strong effectiveness in timeseries imputation againsts SOTA models. The aspect of missing comparison with spatio-temporal models have been raised by other reviewers and stands as an additional limitation of the work.
> >
> > Overall, I do not see space to raise substantially my score.

---

### Official Review · Reviewer_3KyE · 2024-11-04

**Soundness:** 2
**Presentation:** 2
**Contribution:** 3
**Rating:** 5
**Confidence:** 4

**Summary:**

This work describes a new deep reservoir computing architecture (RCN) composed of multiple echo state network (ESN) layers, focused on the task of multivariate time series imputation. The authors then state the echo state property (ESP) for this new network and prove a sufficient condition for the RCN to satisfy ESP. They also study the convergence of RCN, proving an upper limit for the convergence rate. The method is then tested on both synthetic and real-world data.

**Strengths:**

*Originality*

To the best of my knowledge, the architecture introduced in this work (RCN) is original.

*Quality*

The authors perform all the analyses that I would expect from a paper that proposes a new reservoir computing architecture. In particular, they don't just test their method experimentally, they also provide theoretical grounds, rooted in dynamical systems theory, to show that RCN does satisfy ESP and a study of the complexity and efficiency of their method. The experiments are rigorous, with a large selection of baselines to show the network's performance, but also to explore the role of some hyperparameters.

*Clarity*

The claims and the arguments to sustain each of them are logically and consequentially laid out. The structure of the paper follows what one would logically expect.

*Significance*

The topic of how to deal with corrupted data is very relevant. The application of reservoir computing to imputation tasks is relevant, given the efficiency of RC.

**Weaknesses:**

*Quality*

1. In line 161, you state that a monotonically decreasing nonnegative sequence necessarily converges to 0. Maybe you are missing some other assumptions?

2. In line 269, you define the reservoir space as the vector space spanned by the rows of $\boldsymbol R^{(k)}$, so I understand that it's a $(T-\omega)$-dimensional space. I think that this name is a bit misleading since it's usually used to refer to the space of the reservoir states, so the columns of $\boldsymbol R^{(k)}$. Later, in line 279, you mention increasing the number of neurons (i.e. $N$), and you call that the "dimension of the reservoir space". Unless I'm missing something, there is a bit of confusion in terms. But I do find the underlying argument sound.

3. While reading your work, the mind goes quite naturally to the deep ESN architecture described by Gallicchio et al., which you mention in the related works section. There you state that your method is "fundamentally different" form the deep ESN. I do agree that it is different since you have an output layer sandwiched between ESN layers, but I think that more details are required to understand why that structural change is "fundamental". Is it better in terms of performance and efficiency?

4. When proving that RCN has ESP, you never mention (unless I missed it) the standard hypotheses on the compactness of the reservoir and input states. It stands out because the ESP can't be linked only to the dynamics of the reservoir state, it must depend also on the input.

5. In line 431, you state that the MSE results "are lower than 50% of those resulting from other methods", which doesn't seem a fair description of the table for all baselines, in my opinion.

6. One of the claims of the paper is that your method is more efficient, I presume in terms of computational resources. I don't see this claim supported by experiments, could you maybe quantify it?

*Clarity*

There are several mistakes and typos that, to different degrees, affect the readability of the paper. I list here a few:
- At about line 134, the superscripts of the definition of the function $f$ are wrong. Similarly, on line 165, it should be $\boldsymbol{Ar}+\boldsymbol{Bu}\in\mathbb{R}^N$.
- At line 158, it seems to me that the contraction is on the first component of the reservoir state, not the second.
- In line 163, you call "activity function" instead of "activation function". Line 215, "wight" instead of "weight".
- Lines 221-231:
  - The matrix $\boldsymbol C$ I think it should have a superscript $(k)$
  - The definition of $\boldsymbol R^{(k)}$ is a bit confusing since all its components have the same subscript $t$, while it should change from $\omega$ to $T$
  - Sometimes the $\omega$ becomes a $w$. They are very similar, but if you notice the difference it can be confusing.
  - Is the dagger symbol the More-Penrose pseudoinverse?
- The total number of layers is sometimes referred to as $\mathcal K$, sometimes as $K$.
- Line 349, the symbol $p$ was already used to describe to dimension of the input $\boldsymbol u_t$
- Line 373, the reservoir size is referred to as $N_x$, it was just $N$ before
- Line 394, "the algorithm converges in no more than iterations", it's missing something

Apart from typos and other easily fixable mistakes, I report here what are in my opinion some more serious problems:
- The clarity is affected negatively by the use of a quite heavy notation, with a lot of superscripts and subscripts. To mitigate this, maybe something that could help is to remove all the $1:\omega$ subscripts: in my opinion, they are not necessary, especially because they are almost always the same. Would it be possible to just assume $\omega=0$ everywhere and forget about it during the theoretical discussions?
- The parameters $\omega$ (line 221) and reservoir node degree $d$ (line 374) should be defined somewhere first. I presume $\omega$ is the relaxation time, and maybe $d$ is the connectivity (?).

**Questions:**

1. In their work on Deep ESN, Gallicchio et al. argued that a deep structure provides a "multiple time-scales representation" of the input. Is there a similar argument that justifies your modification?

2. Proposition 3.1 provides a sufficient condition for the RCN to have ESP. How does it relate to other criteria developed in the literature? Is it equivalent to the "diagonal Schur stable" condition given by Yildiz et al. in *Re-visiting the echo state property*? It would be nice to expand the relationship between the two, at least to clarify if your proof is also new or if it's standard

3. In "Limitations and future works", you say that ESP is problematic when there is missing data at the beginning of the time series. I don't fully understand the link between ESP and having missing data, could you maybe clarify this?

---

> ### Author Response · Authors · 2024-11-24
>
> # Weakness:
> 1. It also follows from that $\Psi_\Lambda$ is a contraction mapping in r uniformly in u, meaning, there is $L_\Psi<1$ such that $\|\Psi_\Lambda(\mathbf{r},\mathbf{u})-\Psi_\Lambda(\mathbf{s},\mathbf{u})\|\leq L_\Psi\|\mathbf{r}-\mathbf{s}\|$ for all r, s, and u. We have included further derivation details into the revised manuscript, from Line 157 to 161.
> 2. We have changed from "reservoir space" to "space of reservoir states"as suggested. We have further emphasized that $\omega$ is chosen so that $T-\omega>N$ on page 5 of the revised manuscript, and hence the dimension of the space of reservoir states is exact N.
> 3. The fundamental difference between Gallicchio's and our works is two-fold: network structures and training processes. In the deep ESN proposed in Gallicchio's work, the input of the subsequent layer is the reservoir state of the preceding layer, and all the learnable parameterized are trained simultaneously; In our work, the input of the subsequent layer is the "linearly transformed" reservoir state of the preceding layer, meaning, each hidden layer is a complete ESN, and DL-DRCN training is operated layer-by-layer. Conceptually, Gallicchio's deep ESN and our DL-DRCN are analogous to joint learning and independent learning, respectively, where the latter one has better scalability and improved computational efficiency. Moreover, by leveraging ESP, we also prove the convergence of DL-DRCN, which is a general issue to classical independent learning.
> 4. We use the hypothesis that the reservoir map is a contraction map in the reservoir state r uniformly in the input state u, instead of the compactness of the reservoir and input states, to prove ESP. Under our assumption, for each r, the value of $\Psi_\Lambda$ still depends on u, and hence ESP, more specifically the equilibrium point, depends on u as well.
> 5. We have modified the statement and made it more accurate in the attached revised manuscript from Lines 435 to 437.
> 6. Computation time are included in revised Appendix E.
>
> 1,4,5,6,7. Corrected.
> 2. Yes, we only require the reservoir map to be a contraction in the reservoir state. Typos are corrected.
> 3. The notations have been modified. The dagger symbol denotes the transpose of the "regularized Moore-Penrose pseudoinverse", meaning, $A^\dagger=A^{\intercal}(AA^{\intercal}+\beta I)^{-1}$ such that $(A^\dagger)^{\intercal}b={\rm argmin}_{x\in\mathbb{R}^{n}}\big(\|A^{\intercal}a-b\|+\beta\|x\|\big)$, where $A\in\mathbb{R}^{n\times m}$ with $n<m$ and $I$ is the $n\times n$ identity matrix.
>
> 1. We thank the reviewer for this constructive suggestion. However, removing the subscript requires more assumption or the use of expected value notation as the initial transient depends on the choice of random initial reservoir state.
> 2. Corrected.
> # Questions:
> 1. The proposed architecture gives rise to an iterative projection of the time series with missing data to a descending sequence of vector spaces containing the ground truth. Specifically, each ESN layer can be viewed as a projection operator onto the space of its reservoir.
> With the reservoir size decreasing layer-by-layer, the time series to be imputed is iteratively projected to vector spaces of vanishing dimension, which then guarantees its convergence of the ground truth. As mentioned in Gallicchio's deepESN work, adopting different leaky parameters $\alpha$ across different ESN layers enables the framework to achieve multiple time-scales representations, which is definitely applicable to the proposed DL-DRCN as well.
> 2. Proposition 3.1 is new, which generalizes existing conditions for ESP to ESNs with arbitrary activation functions and inputs. Existing references [1-3] mainly focus on the case where activation functions are sigmoid functions, particularly the arctan function, and in [2], the work pointed out by the reviewer, compactness of the spaces of inputs and reservoir states is also required. Specifically, diagonal Schur stability of the weight matrix, together with the arctan function as the activation function, is a special case of the conditions established in Proposition 3.1, those are, the reservoir map is Lipschitz continuous with the Lipschitz constant L and the weight matrix A satisfies $\|A\|<L^{-1}$.
> [1] Michael Buehner et al., "A Tighter Bound for the Echo State Property."
> [2] Izzet B. Yildiz et al., "Re-Visiting the Echo State Property."
> [3] Chenxi Sun et al., "A Review of Designs and Applications of Echo State Networks."
> 3. ESP implies the convergence of the output time series of an ESN as the time step goes to infinity. This provides the theoretical guarantee for the convergence of the output of DL-DRCN to the ground truth time series as the number of ESN layers approaches infinity, as proved in Theorem 4.2. However, data at the beginning of a time series generally do not represent the asymptotic (long time) behavior of the time series so that ESP is not applicable to warrant the imputability of such data.

---

> ### Comment · Reviewer_3KyE · 2024-11-26
>
> Dear authors,
>
> thanks for your replies.
>
> 1. Now it's much better. Just notice that while editing this part, you introduced more typos (missing inequality in line 159, the abrupt start of the sentence in line 162).
> 2. I think there is a misunderstanding on this. I would expect that the reservoir space, or the space of the reservoir states (both names are fine), is the $(N+1)$-dimensional vectorial space where $\tilde{r_t}^{(k)}$ lives. But in line 272, you are using the space spanned by the _rows_ of $\boldsymbol{R}_{\omega:T}^{(k)}$, which live in a $(T-\omega+1)$-dimensional space. I think this can't be called reservoir space, nor "space of the reservoir states", to avoid confusion.
> 3. You say that DL-DRCN "has better scalability and improved computational efficiency" than Gallicchio's deep ESN, but I don't see this statement supported by your analysis or experiments (if I missed it, please point it out to me). As you can see, the other reviewers are also critical of the relationship between the two frameworks, so I think it is crucial to be more thorough about those differences.
> 4,5,6. Thanks for clarifying.
>
> I think almost all the easily fixable concerns that I raised have been addressed, but I think that the main one (maybe I should have stressed it better in my original review) was the relationship with deep ESN. In this regard, I think it is crucial to add experiments to support your claim and justify the introduction of this framework. Therefore, I can't raise the score.

---

### Meta-Review · Area_Chair_HWrZ · 2024-12-17

**Metareview:**

The paper proposes an echo state network (ESN) model for missing data imputation. The model is composed of multiple reservoirs whose readouts are trained layer-wise.

All reviews are negative. Reviewers are especially concerned about the novelty of the method (compared to, e.g., DeepESNs), but the paper also has issues in terms of clarity, presentation, experimental evaluation, and on the validity of the assumptions. The rebuttal was unconvincing for all reviewers.

Overall, I agree with all concerns raised by the reviewers.

**Additional Comments On Reviewer Discussion:**

- **Reviewer CHYy** was concerned about novelty (missing data imputation with ESNs is known) and about the assumptions underlying the method. The rebuttal - in their words - "feels rushed" and they maintained a very negative outlook on the paper.

- **Reviewer A3V4** asked clarifications on missing literature and ablations. In addition, they were concerned there was no real advantage or motivations for using multi-layered ESNs compared to standard ESNs. These concerns remained after rebuttal.

- **Reviewer PnqC** had concerns about lack of baselines and the experimental evaluation. While parts of these concerns were addressed in the rebuttal, the reviewer is still concerned about related works (e.g.,  the DeepESN model) and the presentation.

- **Reviewer XLK9** highlighted many concerns, including lack of novelty (also relative to DeepESN and previous works), unconvincing results, poor presentation, issues in the assumptions, and others. The rebuttal did not address these concerns.

- **Reviewer 3KyE** had many questions on clarity and presentation. Although some were addressed in the rebuttal, the novelty and relation with DeepESNs remain an open problem.

Overall, all reviewers have similar and valid concerns, which have informed my final decision.

---

### Decision · Program_Chairs · 2025-01-22

Reject